# QFree-Det: Query-Free Detector with Transformer and Sequential Matching

## Abstract

Transformer-based detectors, such as DETR and DINO, often struggle with a specific limitation: they can *detect only a fixed number of objects* based on the *predefined* number of queries set. This limitation leads to missed detections when the scene exceeds the model's capacity and increases false positives when the scene contains fewer objects. In addition, existing approaches often combine *one-to-one* and *one-to-many* matching label assignment methods in the decoder for accelerating the model training and convergence. However, this operation introduces a new *detecting ambiguity* issue, which is often overlooked by those methods. To address these challenges, we propose **QFree-Det**, a novel query-free detector capable of dynamically detecting a variable number of objects across different input images. In particular, we present an **A**daptive **F**ree **Q**uery **S**election (AFQS) algorithm to dynamically select queries from the encoder tokens, which efficiently addresses the issue of fixed capacity. Then, we propose a *sequential matching* method that decouples the one-to-one and one-to-many processes into separating sequential steps, effectively addressing the issue of detecting ambiguity. To achieve the sequential matching, we design a new *Location-Deduplication Decoder* (LDD) by rethinking the role of **c**ross-**a**ttention (CA) and **s**elf-**a**ttention (SA) within the decoder. LDD first regresses the location of multiple boxes with CA in a one-to-many manner and then performs object classification to recognize and eliminate duplicate boxes with SA in a one-to-one manner. Finally, to improve the detection ability on small objects, we design a unified PoCoo loss that leverages prior knowledge of box size to encourage the model to pay more attention to small objects. Extensive experiments on COCO2017 and WiderPerson datasets demonstrate the effectiveness of our QFreeDet. For instance, QFree-Det achieves consistent and remarkable improvements over DINO across *five* different backbone models. Notably, QFree-Det obtains a new state-of-the-art of **54.4%** AP and **38.8%** $AP_S$ on `val2017` of COCO with the backbone of VMamba-T under $1\times$ training schedule (**12** epochs), higher than DINO-VMamba-T by **+0.9%** AP and **+2.2%** $AP_S$. The source codes will be released upon acceptance.

## 1 Introduction

In the last few years, *transformer-based* detectors like DEtection TRansformer (DETR) Carion et al. (2020) and DINO Zhang et al. (2022b), have simplified the detection pipeline by providing end-to-end detection capabilities and demonstrating promising performance in comparison to classical CNN-based detectors Girshick (2015); Ren et al. (2015); Liu et al. (2016); Redmon et al. (2016); He et al. (2017). Subsequently, a series of follow-up works have been proposed to boost DETR on the architecture of encoder and decoder Zhu et al. (2020); Cao et al. (2022), query formulations Meng et al. (2021); Liu et al. (2022); Cao et al. (2024), and training efficacy Meng et al. (2021); Zhang et al. (2022a); Li et al. (2022); Zhang et al. (2022b); Jia et al. (2023); Hu et al. (2024).

**DETR-like models: the limited number of objects detection (OD) dilemma.** DETR-like models Carion et al. (2020); Meng et al. (2021); Yao et al. (2021); Liu et al. (2022); Li et al. (2022); Zhang et al. (2022b); Zheng et al. (2023); Cao et al. (2024) utilize a transformer encoder-decoder architecture to treat OD as a set prediction problem. These models make predictions based on a *fixed-size* set of $N$ learnable object queries, and *each query is responsible for predicting a single object*, where

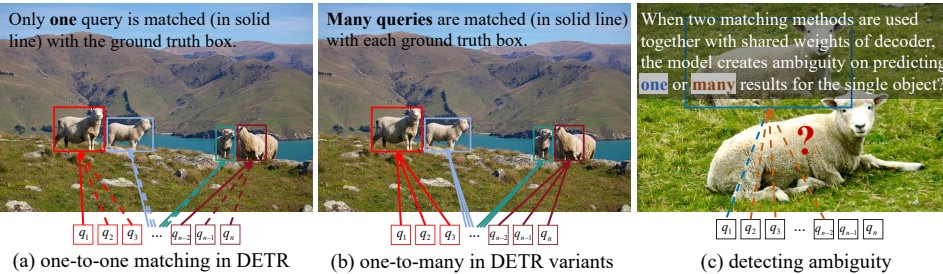

(a) one-to-one matching in DETR    (b) one-to-many in DETR variants    (c) detecting ambiguity

Figure 1: The mixing of one-to-one and one-to-many matching label alignment approaches with shared weights in decoder introduces detecting ambiguity in predicting one or many results for a single object. This ambiguity often leads the model to predict multiple detection results for the same object, increasing false positives.

these predicted objects are matched with ground truth objects following a *one-to-one* matching label assignment manner. However, this approach imposes a notable dilemma: the model can detect only "*limited number of objects*", as the fixed size of $N$ queries becomes a hyper-parameter tied to the model's weight, which significantly hampers the model's flexibility and applicability when dealing with input images containing objects more than $N$. This dilemma raises the natural question: *is there an end-to-end approach that can predict a free number of objects and surpass the performance of state-of-the-art (SOTA) DINO?* We term this issue as the "*free-object predictions*" problem.

**Detecting ambiguity.** One-to-one matching is a fundamental design feature of DETR-like models, enabling their end-to-end capability in OD without the need for a post-process like non-maximum suppression (NMS) to remove duplicate detections. To address the low training efficacy and slow convergence speed of the model, many works Carion et al. (2020); Zhang et al. (2022b); Jia et al. (2023); Hu et al. (2024) adopt the one-to-many matching approach to increase the positive samples with an *auxiliary* decoder or branches. However, the one-to-one and one-to-many matching label alignment methods are *mutually exclusive*. They operate on the same shared decoder weights, which introduces ambiguity in the model: it becomes unclear whether the model should predict one or multiple results for the same object, as illustrated in Fig. 1 (c).

**QFree-Det.** To achieve the "free-object predictions" and address the "detecting ambiguity" issue, we develop a novel end-to-end transformer-based *query-free* detector termed **QFree-Det**, which frees the constraint on the fixed number of queries, allowing the model to detect adaptive quantities of objects for any given input images. Our contribution can be summarized as follows: **(1)** We present a new AFQS algorithm to dynamically select a flexible number of queries from the encoder tokens, which solves the fixed capacity issue. **(2)** We introduce a new *sequential matching* method that separates the one-to-one and one-to-many processes into two independent parts, effectively addressing the issue of detecting ambiguity. To achieve this, we rethink the roles of core modules in the decoder for predicting object bounding boxes and classes: *cross-attention* and *self-attention*, and further design a new end-to-end *Location-Deduplication Decoder* (LDD), which decomposes the detection process into two simple steps: **boxes locating** and **objects deduplication**. Specifically, the LDD includes two parts of **B**ox **L**ocating **P**art (BLP) and **D**eduplication **P**art (DP), respectively. The BLP aims to accurately locate the extensive bounding boxes layer-by-layer for potential objects and improve the training and convergence efficiency with one-to-many matching; in contrast, the DP is designed for removing duplicate boxes through classification supervision with one-to-one matching. Our model does not require additional decoder branches, maintains training efficiency, and achieves "free-object predictions", streamlining the model architecture. **(3)** We design a new loss function, which incor**PO**rates **C**lassification with I**O**u and B**O**x Size (PoCoo) for re-weighting the classification loss, to improve the detection ability on small objects. **(4)** Extensive experiments on COCO2017 and WiderPerson datasets demonstrate that QFree-Det achieves promising performance compared with many existing methods across many different backbone models.

## 2 RELATED WORKS

**Transformer detectors.** DETR Carion et al. (2020), as the pioneer transformer-based detector, represents a significant breakthrough in OD. By framing OD as a direct set prediction task and leveraging transformer architectures, DETR introduces a more efficient end-to-end detection paradigm

while eliminating the need for hand-designed components like NMS. Since then, many follow-up efforts have focused on various aspects of DETR enhancement, including accelerating the model training Meng et al. (2021); Li et al. (2022); Zhang et al. (2022b); Jia et al. (2023); Zong et al. (2023), reformulating the decoder queries Meng et al. (2021); Yao et al. (2021); Liu et al. (2022); Zhang et al. (2023), improving the encoder and decoder architectures Zhu et al. (2020); Roh et al. (2021); Cao et al. (2022), and optimizing loss functions Liu et al. (2023a); Cai et al. (2023); Pu et al. (2024); Hu et al. (2024). While these methods effectively enhance the detection capabilities, they often overlook the vital dilemma of "*limited number of objects detecting*", stemming from DETR's single-shot approach to aligning queries with objects via bipartite matching.

**Free-form object detection.** In real-world scenarios, the number of detectable objects varies widely Shao et al. (2018); Zhang et al. (2019), ranging from individual instances to thousands, presenting significant challenges for detectors Liu et al. (2021a); Cheng et al. (2023). The "free-object predictions" problem in DETR-like models presents two challenges: when the number of objects to be detected is much larger than the predefined queries, the model will miss detections; in contrast, when the number of objects to be detected is far fewer than the predefined queries, the model would introduce a substantial amount of redundant computation, leading to an increased false positive rate and a decrease in detection performance.

Recently, a diffusion-based model called DiffusionDet Chen et al. (2023b) introduced a novel framework that formulates OD as a denoising diffusion process from numerous noisy boxes to refined object boxes, achieving the flexibility to predict an arbitrary number of detections by decoupling training and evaluation processes and leveraging iterative evaluation. However, despite its advancements, DiffusionDet still suffers from several limitations. Notably, it requires hand-designed postprocessing with NMS for duplicate box removal, complicating both the training and inference process; additionally, its adaptability is constrained by manually defined parameters of noisy box number and evaluation iterations, increasing the evaluation complexity; moreover, DiffusionDet lags behind the SOTA works such as DINO Zhang et al. (2022b), hindering its potential for development and application.

**One-to-one matching.** DETR Carion et al. (2020) and its variants, such as Deformable DETR Zhu et al. (2020) and DAB-DETR Liu et al. (2022), innovate with a one-to-one set matching approach for end-to-end object detection, as shown in Fig. 1 (a), bypassing the need for conventional hand-crafted NMS to remove duplicate detections. Though streamlining the detection workflow, this one-to-one matching manner leads to *only a few queries assigned as positive samples*, thereby significantly diminishing the training efficiency of positive samples due to sparse supervision. **One-to-many matching.** To address the limitations of one-to-one matching and boost training efficiency, many efforts, including Hybrid-DETR Jia et al. (2023), Co-DETR Zong et al. (2023), Group-DETR Chen et al. (2023a), Align-DETR Cai et al. (2023) and DAC-DETR Hu et al. (2024), etc, have explored one-to-many label assignments for increasing the matched positive samples among dense queries. By explicitly assigning multiple queries to each ground truth box, these methods boost the quantity of positive matches, accelerate model convergence, and enhance training efficiency.

Nevertheless, the shift towards one-to-many matching naturally introduces a new *"detecting ambiguity"* issue, which contradicts the foundational one-to-one principle of DETR. This ambiguity arises when mixing the one-to-many and one-to-one matching schemes through shared weights in auxiliary decoders or branches. It creates uncertainty about whether one or multiple results should be predicted for a single object, as depicted in Fig. 1 (c). Unfortunately, this issue is overlooked in existing works Carion et al. (2020); Zhang et al. (2022b); Jia et al. (2023); Cai et al. (2023); Hu et al. (2024). Furthermore, the adoption of *additional* decoder branches to enable one-to-many matching, as observed in Hybrid-DETR Jia et al. (2023) and DAC-DETR Hu et al. (2024), significantly increases training complexity and costs, further exacerbating the ambiguity.

## 3 METHODS

In this section, we first address the challenges by rethinking the main pipeline, model composition, and the roles of each component of DETR-like models. We then introduce the overall architecture of QFree-Det and propose the novel AFQS algorithm to achieve "free-object predictions". Furthermore, we present our sequential matching approach to eliminate the detecting ambiguity through our innovative LDD framework. Finally, we present PoCoo loss for improving small object detection.

## 3.1 OVERVIEW OF DETR-LIKE FRAMEWORK

### 3.1.1 MAIN PIPELINE OF DETR-LIKE MODELS.

The DETR-like architecture comprises three main modules: a compact backbone for feature extraction, a transformer encoder neck for feature enhancement, and a transformer-decoder for predicting bounding boxes and classes. Given an input image $I \in \mathbb{R}^{H \times W \times 3}$ (H, W: image height and width), the backbone extracts a compact feature representation $B$. This feature $B$ is then passed through the transformer encoder, which consists of a chain of attention Dosovitskiy et al. (2020) or deformable attention Zhu et al. (2020) layers to disentangle objects and obtain the encoder feature **E**. Next, the model initializes a fixed-size set of two types of queries: content query (CQ) and positional query (PQ) Liu et al. (2022); Zhang et al. (2022b). These queries, along with the encoder feature **E**, are fed into the transformer decoder, which updates these queries based on the information from the encoder feature via the SA and CA modules. Finally, the content queries are passed through two separate **f**eed **f**orward **n**etworks (FFN) to predict the bounding box coordinates and class labels, respectively. While the aforementioned framework achieves end-to-end detection and demonstrates promising detection performance, the specific roles of various components in the model, such as the meaning of CQ and PQ, the effects of SA and CA, remain unclear, which poses a limitation to the further development of the model.

### 3.1.2 RETHINKING THE ROLE OF CONTENT QUERY AND POSITIONAL QUERY

CQ and PQ are important modules in the DETR-like framework for OD. These modules have been optimized in a series of research works Zhu et al. (2020); Meng et al. (2021); Yao et al. (2021); Wang et al. (2022); Liu et al. (2022); Zhang et al. (2022b), as detailed in the **Appendix** (Sec.A.1). However, the issue of "*free-object prediction*" is *directly limited by the fixed number of CQ and PQ*, and the actual roles of these two types of queries in the detection process need to be further explored. The **CQ**, also known as the decoder embeddings in DETR, plays a clear role, which is *indispensable* for OD to predict the bounding boxes and classes of objects directly. In the SA and CA modules of the decoder, the **PQ** is added to the CQ to increase the variation of CQ Carion et al. (2020) and facilitate predicting diverse detection results. However, the exact role of PQ is not yet clear. To gain further insights, we conduct ablation experiments with DINO under two different backbones on COCO Lin et al. (2014), removing the PQ from these two modules and assessing its impact. As illustrated in Table 1, the ablation results reveal that the removal of the PQ has a minimal effect on the

Table 1: The ablation on PQ.

| Backbone | PQ in SA | PQ in CA | AP |
|---|---|---|---|
| ResNet50 He et al. (2016) | ✓ | ✓ | 49.5 |
| | ✓ | ✗ | 49.8 (+0.3) |
| | ✗ | ✓ | 49.3 (-0.2) |
| | ✗ | ✗ | 49.3 (-0.2) |
| Strip-MLP-T Cao et al. (2023) | ✓ | ✓ | 51.7 |
| | ✓ | ✗ | 51.6 (-0.1) |
| | ✗ | ✓ | 51.8 (+0.1) |
| | ✗ | ✗ | 51.7 |

model's performance, with only a variation of $\pm$**0.3%** in average precision (AP). In response to the requirement for "free-object predictions", we directly remove the PQ and propose a novel query type termed the self-adaption decoder query (SADQ), which further replaces the fixed number of static embeddings of CQ, enabling the model to generate a flexible number of predictions. The algorithm for constructing this adaptive query is outlined in Sec 3.3 of the proposed AFQS algorithm.

### 3.1.3 SELF-ATTENTION AND CROSS-ATTENTION

**Self-attention and its high GPU memory demand.** In the decoder of DETR-like models, the SA layer operates on a fixed number of $N$ queries $Q = \{q_1, ..., q_n\}$ as the input of both query, key, and value for the attention process. The self-attention map is computed using the formulation $Attention(Q, K, V) = softmax(\frac{QK^T}{\sqrt{d_k}})V$, where $d_k$ is the scaling factor. To obtain attention scores for all *pairwise* queries, the operation $QK^T$ constructs a matrix with size $n \times n$, resulting in a complexity of $O(n^2)$. However, this issue of quadratic complexity becomes particularly significant for free-form object detection. Specifically, SA may need to construct larger matrices of arbitrary sizes, such as $20,000 \times 20,000$ or even larger. As the number of queries increases, the model would present a new challenge regarding GPU resources and memory requirements, making it difficult to *train* the model. Accordingly, the natural question comes: *what is the role of SA? Can SA be removed directly to solve this problem?*

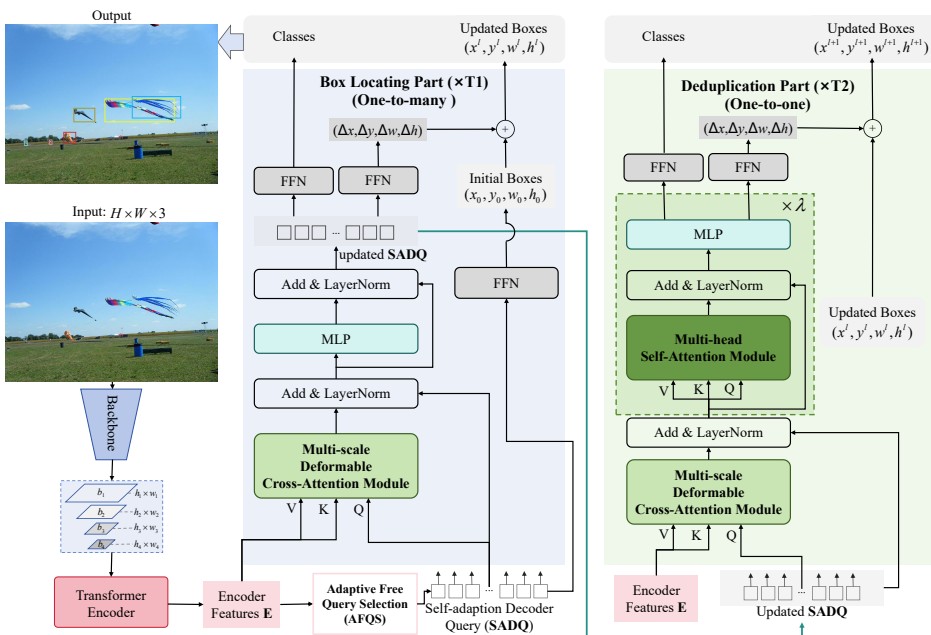

Figure 2: The overall architecture of the QFree-Det model.

**Role and necessity of SA in transformer decoder.** To answer these two questions, we conduct ablations on SA with the DINO model on COCO Lin et al. (2014) in Table 2. Directly removing SA leads to a substantial **4.9%** drop in AP, while the decrease in average recall (AR) is only **0.6%**. This phenomenon suggests that *SA plays a crucial role in reducing false positive predictions*. To address the "free-object predictions" problem, we introduce a new variant termed the "Binary" model, which adds a binary classification branch connected to the encoder. This branch dynamically selects queries from encoder tokens based on the binary classification result. This "Binary" model achieves a **+2.7%** AP improvement over the baseline. Further incorporating NMS leads to an additional **+0.7%** AP

Table 2: Ablation on SA layers. The QNum means the number of queries is fixed or not. $AP_S$ and $AP_L$ are the AP for small and large objects, respectively.

| Method | SA | QNum | AP | $AP_S$ | $AP_L$ | AR | $AR_S$ | $AR_L$ |
|---|---|---|---|---|---|---|---|---|
| Baseline | ✓ | fixed | 49.0 | 32.0 | 63.0 | 72.7 | 55.9 | 88.4 |
| | ✗ | fixed | 44.1(-4.9) | 27.9 | 56.7 | 72.1(-0.6) | 53.3 | 86.6 |
| Binary | ✗ | *free* | 46.8 | 29.9 | 59.7 | 71.9 | 53.6 | 87.5 |
| Binary-NMS | ✗ | *free* | 47.5 | 30.5 | 61.3 | 69.3 | 53.9 | 83.3 |
| Binary-Linear-Attn | ✓ | *free* | 48.4 | 30.4 | 62.4 | 73.9 | 58.9 | 88.0 |
| Binary-SA | ✓ | *free* | 49.1 | 31.4 | 63.4 | 73.1 | 57.9 | 88.3 |

gain, suggesting that SA module provides a similar performance boost as NMS. To reduce the quadratic complexity of SA, we adopt **linear transformer** architectures, such as efficient attention Shen et al. (2021) and external attention Guo et al. (2023), which avoid computing *pairwise* attention scores for each input token, reducing the complexity. We construct a new variant of Binary-Linear-Attn, allowing for a more flexible number of query selections. However, as presented in Table 2, the experiment result shows that Binary-SA significantly outperforms Binary-Linear-Attn, suggesting that *computing attention maps between all query pairs is a suitable approach for removing duplicate predictions*. These results consistently highlight the significance and necessity of SA for eliminating duplicate detections, which inspired us to design the DP for LDD in Sec. 3.4. We argue that duplications are removed by the SA via computing similarities between each paired query and further updating queries through one-to-one matching and class supervision in training process. Please refer to the Sec. A.2 of **Appendix** for more detailed information.

**Opposing role of CA and SA.** The majority of existing DETR-like *one-to-many* decoder architectures Jia et al. (2023); Zong et al. (2023); Cai et al. (2023); Hu et al. (2024) suffer from the problem of detecting ambiguity. This issue is also aroused by the opposing impacts on the object queries of CA and SA, where the SA disperses queries from each other, and the CA tends to gather multiple queries around the same object Hu et al. (2024). In our analysis, CA primarily updates decoder queries by performing cross-attention to flow the object information from the encoder feature to decoder queries. This results in multiple queries being linked to the same object for more accurate

bounding box predictions, namely the one-to-many process. While this process increases the accuracy of bounding box detections, it inevitably leads to a new issue of *increasing the false positive detections*. Most existing decoder architectures apply the one-to-one process of SA before the one-to-many process of CA layer-by-layer. This *recurrent shifting* between the two opposing processes leads to the significant detecting ambiguity problem.

Based on the above analysis, we propose a new LDD framework in Sec. 3.4, which decouples the detection process into two simple parts: box locating with CA and duplicate detections removing with SA. LDD mitigates the detecting ambiguity while retaining the benefits of CA and SA.

## 3.2 MAIN ARCHITECTURE OF QFREE-DET

Fig. 2 presents an overview of the novel QFree-Det model. The input image $I \in \mathbb{R}^{H \times W \times 3}$ is processed by the backbone and encoder to obtain the enhanced feature representations $\mathbf{E}$. Then, the proposed algorithm AFQS operates on all encoder tokens to generate a variable number of self-adaptive decoder queries. These decoder queries, along with the encoder features $\mathbf{E}$, are input to the **Box Locating Part** (BLP) for $T_1$ iterations, aiming to locate the bounding box of each object with multiple queries and keep the training efficacy through one-to-many matching. Subsequently, the **Deduplication Part** (DP), consisting of $T_2$ iterations, takes the feature $\mathbf{E}$ and updated decoder queries as input to remove duplicate detections by one-to-one matching. Each decoder layer produces the bounding box and classification results for the detected objects. The initial locations of bounding boxes $\boldsymbol{B}_{x_0 y_0 w_0 h_0}$ are initially predicted by the encoder and then refined layer-by-layer in the decoder by regressing the box offsets. Mathematically, the whole process can be represented as:

$$\boldsymbol{B}_{x_0 y_0 w_0 h_0} = FFN(AFQS(\boldsymbol{E})) \tag{1}$$

$$\boldsymbol{B}_{xywh}^i = \boldsymbol{B}_{x_0 y_0 w_0 h_0} + \sum_{i=1}^{T_1} BLP(\boldsymbol{q}_d^i, \boldsymbol{E}) + \sum_{\substack{j=i-T_1 \\ i > T_1}}^{T_2} DP(\boldsymbol{q}_d^j, \boldsymbol{E}) \tag{2}$$

where $\boldsymbol{q}_d$ denotes the feature of the self-adaption decoder query; $i$ and $j$ indexes the layer of $\boldsymbol{q}_d$.

## 3.3 ADAPTIVE FREE QUERY SELECTION ALGORITHM

Recall that we have analyzed the role of CQ and PQ in Sec. 3.1.2. To construct the query-free detection framework, we eliminate the unnecessary PQ and introduce the new AFQS algorithm. This algorithm dynamically obtains appropriate decoder queries tailored to different images, allowing for generating a flexible number of detections.

**Shifting the fixed number of queries into dynamic initialization.** In existing models Carion et al. (2020); Zhu et al. (2020); Li et al. (2022); Zhang et al. (2022b); Liu et al. (2023a), CQ is designed as a fixed number of queries, either with learnable static queries or initialized with zero vectors. However, these fixed queries limit the model's flexibility, and the static initialization hinders the model from dynamically adapting to different images. Consequently, these limitations significantly restrict the models' detection capabilities. To address these issues and achieve query-free detection, we introduce our new self-adaptive decoder query (SADQ) to replace CQ. SADQ is obtained by sorting the classification scores of all $N$ encoder tokens and selecting the scores above a certain threshold $S$ as $M$ SADQ. This **t**hreshold method with **s**orting **c**lassification **s**core (TSCS) approach enables the

---

**Algorithm 1** Adaptive free query selection

**Input:** token sequence of encoder $\mathbf{E}$ : (B, N, D)
**Output:** decoder query $\mathbf{SADQ}$ : (B, M, D)
  **Initialize:**
    $S \leftarrow s \in (0, 1)$; ▷threshold of classification score
    $P \leftarrow p \in (0, N)$; ▷the size of query pool
    $\mathbf{T}_{enc}^{class}$ : (B, N, C) $\leftarrow$ **FFN**($\mathbf{E}$)
    $\mathbf{T}_{idx}$ : (B, M) $\leftarrow$ **filter** ($\mathbf{T}_{enc}^{class}, S$)
  /* select queries in training mode */
  **if** $training$ and $M \leq P$ **then**
    $\mathbf{SADQ}$ : (B, M, D) $\leftarrow$ **index**($\mathbf{E}, \mathbf{T}_{idx}$)
  **else if** $training$ and $M > P$ **then**
    $\mathbf{T}_{idx}$ : (B, P) $\leftarrow$ **TopK** ($\mathbf{T}_{enc}^{class}, P$)
    $\mathbf{SADQ}$ : (B, P, D) $\leftarrow$ **index**($\mathbf{E}, \mathbf{T}_{idx}$)
  **else if** $testing$ **then**
    $\mathbf{SADQ}$ : (B, M, D) $\leftarrow$ **index**($\mathbf{E}, \mathbf{T}_{idx}$)
  **end if**
  **return** $\mathbf{SADQ}$

---

model to adaptively select a variable number of queries for different input images. However, ensuring that the same number of queries for different images is used within the same batch for practical training is necessary during the training process. To overcome this challenge, we utilize an "Alignment Approach", which is detailed in the Sec. A.3 of **Appendix**.

**Approach for saving GPU memory demand.** To tackle the large GPU resources and memory challenge for SA (in Sec. 3.1.3), we define a pool with size $P$ during the training stage. When the number of selected queries is higher than $P$, the TopK method is used to select $P$ queries from $M$ queries to remove the duplicate queries. During the *early training stages*, the classification scores of most encoder tokens are randomly distributed, leading to many redundant queries being selected. As training progresses, the classification scores increase for selected queries and decrease for others, leading to a decrease in query redundancy. The algorithm AFQS is illustrated in Algorithm 1. The function of **filter** means the TSCS method. The **index** function aims to dynamically select SADQ from $\mathbf{E}$ based on the index of token $\mathbf{T}_{idx}$.

## 3.4 LOCATION-DEDUPLICATION DECODER

To effectively mitigate detecting ambiguity while maintaining training efficacy, we design a novel LDD framework by decoupling the one-to-one and one-to-many matching processes. The LDD comprises two parts: one for locating the object and the other for removing duplicate detections.

**Box Locating Part.** The BLP is specifically designed to accurately locate more potential objects using multiple queries following a one-to-many matching manner, where the one-to-many matching is achieved by repeating the ground truth box for $K$ times and aligning each repeated box with one query. Considering the different roles of CA in gathering multiple queries around the same object and SA in removing duplicate detections, we only utilize CA in this part for object prediction. Mathematically, this process can be formulated as:

$$\text{BLP}(\boldsymbol{q}_d, \boldsymbol{E}) = LN\{MLP(LN(\boldsymbol{q}_d + CA(\boldsymbol{q}_d, \boldsymbol{E}))) + LN(\boldsymbol{q}_d + CA(\boldsymbol{q}_d, \boldsymbol{E}))\} \tag{3}$$

where LN denotes the LayerNorm Ba et al. (2016) layer. The MLP consists of two linear layers to facilitate the interaction of channel information. $CA$ means the cross-attention operation through the multi-scale deformable attention Zhu et al. (2020).

**Deduplication Part.** While the BLP achieves accurate object localization by assigning multiple queries to the same object, it also introduces many false positive queries. To address these duplicate queries, we introduce the Deduplication Part (DP), which performs one-to-one matching to remove redundancies. The DP consists of two components: the $\text{DP}_{\text{CA}}$ module, which refines the box location using the CA with one-to-one matching form, and the multi-head self-attention block (MSAB), which combines SA and MLP to remove duplicate detections. The SA module and one-to-one label alignment is combined together in DP to achieve one-to-one matching, and we iterate this block for $\lambda$ times to enhance the deduplication process. The overall process can be described as follows:

$$\text{DP}_{\text{CA}}(\boldsymbol{q}_d, \boldsymbol{E}) = LN(\boldsymbol{q}_d + CA(\boldsymbol{q}_d, \boldsymbol{E})) \tag{4}$$

$$\text{DP}_{\text{MSAB}}(\boldsymbol{q}_d, \boldsymbol{E}) = \sum_{m=1}^{\lambda} MLP^m\{LN^m\left[\boldsymbol{q}_d^{m-1} + SA^m(\boldsymbol{q}_d^{m-1})\right]\} \tag{5}$$

where $\lambda$ indexes the block number. $SA$ denotes the multi-head self-attention module Vaswani et al. (2017). The $\boldsymbol{q}_d^0$ is obtained by $\text{DP}_{\text{CA}}$, and $\boldsymbol{q}_d^m$ is updated by $\text{DP}_{\text{MSAB}}$. The MSAB block significantly helps the model to reduce false positive detections by using multiple SA, demonstrated by the ablation in Table 6. Specifically, during the training process, by computing the pairwise attention score between all queries, the assigned query for each ground truth object gradually obtains a high classification score. In contrast, the remaining queries associated with the same ground truth receive lower scores through one-to-one matching and classification supervision.

**Stop gradient back-propagation from DP to BLP of queries.** Since the query is sequentially updated by the BLP and DP modules, the one-to-many matched queries in BLP may be matched in a one-to-one fashion in DP. This can result in conflicting supervision and chaotic gradient updates, leading to a re-emergence of detection ambiguity. To address this issue, we take an additional step by **s**topping the **g**radient back-propagation of the **q**ueries (SGQ) from the DP to the BLP during training, ensuring consistent matching across different parts, as demonstrated in Table 16.

## 3.5 CLASSIFICATION LOSS WITH IOU AND BOX SIZE

To address the misalignment of queries between the classification score and box regression result, Align-DETR Cai et al. (2023) introduced an IA-BCE loss by combining Iou and predicted classification score as new label $t$ in binary cross entropy (BCE) loss to align these two scores. To

Table 3: Comparison with previous popular detectors on `val2017` of COCO. The FLOPs of QFree-Det are calculated on a 1280×800 resolution with 900 queries, which matches the configuration of the baseline DINO.

| Model | Year | Backbone | Objects | Epochs | AP | AP$_{50}$ | AP$_{75}$ | AP$_S$ | AP$_M$ | AP$_L$ | Params | FLOPs |
|---|---|---|---|---|---|---|---|---|---|---|---|---|
| DETR (Carion et al., 2020) | 2020 | ResNet50 | fixed | 500 | 42.0 | 62.4 | 44.2 | 20.5 | 45.8 | 61.1 | 41M | 86G |
| DETR-DC5 (Carion et al., 2020) | 2020 | ResNet50 | fixed | 500 | 43.3 | 63.1 | 45.9 | 22.5 | 47.3 | 61.1 | 41M | 187G |
| Deformable-DETR (Zhu et al., 2020) | 2020 | ResNet50 | fixed | 50 | 46.2 | 65.2 | 50.0 | 28.8 | 49.2 | 61.7 | 40M | 173G |
| Conditional DETR (Meng et al., 2021) | 2021 | ResNet50 | fixed | 108 | 43.0 | 64.0 | 45.7 | 22.7 | 46.7 | 61.5 | 44M | 90G |
| Sparse-DETR (Roh et al., 2021) | 2021 | ResNet50 | fixed | 50 | 46.3 | 66.0 | 50.1 | 29.0 | 49.5 | 60.8 | 41M | 136G |
| DAB-DETR (Liu et al., 2022) | 2022 | ResNet50 | fixed | 50 | 42.6 | 63.2 | 45.6 | 21.8 | 46.2 | 61.1 | 44M | 100G |
| DN-DETR (Li et al., 2022) | 2022 | ResNet50 | fixed | 50 | 44.1 | 64.4 | 46.7 | 22.9 | 48.0 | 63.4 | 44M | 94G |
| Efficient-DETR (Yao et al., 2021) | 2021 | ResNet50 | fixed | 36 | 44.2 | 62.2 | 48.0 | 28.4 | 47.5 | 56.6 | 32M | 159G |
| CF-DETR (Cao et al., 2022) | 2022 | ResNet50 | fixed | 36 | 47.8 | 66.5 | 52.4 | 31.2 | 50.6 | 62.8 | - | - |
| Focus-DETR (Zheng et al., 2023) | 2023 | ResNet50 | fixed | 36 | 50.4 | 68.5 | 55.0 | 34.0 | 53.5 | 64.4 | 48M | 154G |
| DiffusionDet (Chen et al., 2023b) | 2023 | ResNet50 | *free* | 60 | 46.8 | 65.3 | 51.8 | 29.6 | 49.3 | 62.2 | - | - |
| Grounding DINO (Liu et al., 2023b) | 2023 | ResNet50 | fixed | 12 | 48.1 | 65.8 | 52.3 | 30.4 | 51.3 | 62.3 | - | - |
| Co-DETR-4scale Zong et al. (2023) | 2023 | ResNet50 | fixed | 12 | 49.5 | 67.6 | 54.3 | 32.4 | 52.7 | 63.7 | - | - |
| Stable-DINO Liu et al. (2023a) | 2023 | ResNet50 | fixed | 12 | 50.4 | 67.4 | 55.0 | 32.9 | 54.0 | 65.5 | 47M | 279G |
| DETA Ouyang-Zhang et al. (2022) | 2022 | ResNet50 | fixed | 12 | 50.5 | 67.6 | 55.3 | 33.1 | 54.7 | 65.2 | 52M | - |
| DDQ DETR Zhang et al. (2023) | 2023 | ResNet50 | fixed | 12 | 51.3 | 68.6 | 56.4 | 33.5 | 54.9 | 65.9 | - | - |
| MS-DETR Zhao et al. (2024) | 2024 | ResNet50 | fixed | 12 | 50.3 | 67.4 | 55.1 | 32.7 | 54.0 | 64.6 | - | - |
| Align-DETR Cai et al. (2023) | 2023 | ResNet50 | fixed | 12 | 50.2 | 67.8 | 54.4 | 32.9 | 53.3 | 65.0 | 47M | 279G |
| DAC-DETR Hu et al. (2024) | 2024 | ResNet50 | fixed | 12 | 50.0 | 67.6 | 54.7 | 32.9 | 53.1 | 64.2 | - | - |
| DINO (Zhang et al., 2022b) | 2022 | ResNet50 | fixed | 12 | 49.0 | 66.6 | 53.5 | 32.0 | 52.3 | 63.0 | 47M | 279G |
| **QFree-Det (ours)** | 2024 | ResNet50 | *free* | 12 | 50.5 +(1.5) | 67.5 | 55.1 | **34.3 (+2.3)** | 54.6 | 64.5 | 48M | 275G |
| Align-DETR Cai et al. (2023) | 2023 | ResNet50 | fixed | 24 | 51.3 | 68.2 | 56.1 | 35.5 | 55.1 | 65.6 | 47M | 279G |
| DAC-DETR Hu et al. (2024) | 2024 | ResNet50 | fixed | 24 | 51.2 | 68.9 | 56.0 | 34.0 | 54.6 | 65.4 - | - | |
| DINO (Zhang et al., 2022b) | 2022 | ResNet50 | fixed | 24 | 50.4 | 68.3 | 54.8 | 33.3 | 53.7 | 64.8 | 47M | 279G |
| DINO (Zhang et al., 2022b) | 2022 | ResNet50 | fixed | 36 | 50.9 | 69.0 | 55.3 | 34.6 | 54.1 | 64.6 | 47M | 279G |
| **QFree-Det (ours)** | 2024 | ResNet50 | *free* | 24 | **51.3 (+0.9)** | 68.4 | 55.9 | **35.5 (+2.2)** | 54.8 | 65.7 | 48M | 275G |
| DINO Zhang et al. (2022b) | 2022 | Strip-MLP-T | fixed | 12 | 51.7 | 69.5 | 56.8 | 34.7 | 55.1 | 66.0 | 44M | 263G |
| **QFree-Det (ours)** | 2024 | Strip-MLP-T | *free* | 12 | 52.8 (+1.1) | 70.1 | 57.6 | 35.7 (+1.0) | 56.8 | 67.4 | 45M | 264G |
| **QFree-Det (ours)** | 2024 | Strip-MLP-T | *free* | 24 | 54.5 | 72.0 | 59.5 | 37.4 | 58.5 | 69.6 | 45M | 264G |
| **QFree-Det (ours)** | 2024 | Strip-MLP-T | *free* | 36 | 55.0 | 72.5 | 60.0 | 38.4 | 58.9 | 69.9 | 45M | 264G |
| $\mathcal{H}$-Deformable-DETR (Jia et al., 2023) | 2023 | Swin-T | fixed | 12 | 50.6 | 68.9 | 55.1 | 33.4 | 53.7 | 65.9 | - | - |
| $\mathcal{H}$-Deformable-DETR (Jia et al., 2023) | 2023 | Swin-T | fixed | 36 | 53.2 | 71.5 | 58.2 | 35.9 | 56.4 | 68.2 | - | - |
| DINO (Ren et al., 2023) | 2023 | Swin-T | fixed | 12 | 51.3 | 69.0 | 56.0 | 34.5 | 54.4 | 66.0 | 48M | 280G |
| **QFree-Det (ours)** | 2024 | Swin-T | *free* | 12 | 52.7 (+1.4) | 70.2 | 57.6 | 36.3 (+1.8) | 56.4 | 67.7 | 49M | 281G |
| **QFree-Det (ours)** | 2024 | Swin-T | *free* | 24 | 54.4 | 71.9 | 59.3 | 38.1 | 58.1 | 69.2 | 49M | 281G |
| **QFree-Det (ours)** | 2024 | Swin-T | *free* | 36 | **54.9** | **72.4** | **59.9** | **38.3** | **58.6** | **69.6** | 49M | 281G |
| DINO Zhang et al. (2022b) | 2022 | Swin-L | fixed | 12 | 56.8 | 75.6 | 62.0 | 40.0 | 60.5 | 73.2 | 218M | 945G |
| **QFree-Det (ours)** | 2024 | Swin-L | *free* | 12 | **57.7 (+0.9)** | 75.6 | **63.0** | 40.0 | **62.5** | **74.2** | 219M | 946G |
| DINO (Zhang et al., 2022b) | 2022 | Swin-L | fixed | 36 | 58.0 | 76.1 | **64.0** | 40.1 | 62.2 | **74.3** | 218M | 945G |
| **QFree-Det (ours)** | 2024 | Swin-L | *free* | 24 | **58.2** | **76.1** | 63.6 | **41.6** | **62.7** | 74.2 | 219M | 946G |
| DINO | 2024 | VMamba-T | fixed | 12 | 53.5 | 71.5 | 58.2 | 36.6 | 56.7 | 68.3 | 50M | 290G |
| **QFree-Det (ours)** | 2024 | VMamba-T | *free* | 12 | **54.4 (+0.9)** | **72.0** | **59.2** | **38.8 (+2.2)** | **58.2** | **69.2** | 51M | 290G |

further improve the detection ability of small objects, we develop a new unified PoCoo loss that incor**PO**rates **C**lassification with I**O**u and B**O**x Size:

$$
\text{PoCoo} = \sum_i^{N_{pos}} BCE(p_i, t_i) \times \left[ \left( 1 - \sqrt{\frac{h_i}{H} \frac{w_i}{W}} \right)^{\alpha} + 1 \right] + \sum_j^{N_{neg}} p_j^2 BCE(p_j, 0) \tag{6}
$$

where $i$ and $j$ indexes the prediction of objects, $h_i$ and $w_i$ denote the height and width of the matched ground truth box. $p$ and $t$ represent the predicted classification score and new label, respectively. $\alpha$ ranges between 0 and 1. The distinction between our PoCoo loss and IA-BCE loss lies in the $[*]$ term. In Eq 6, we introduce the prior of box size information into the loss function, explicitly assigning higher weights to small objects and encouraging the model to pay more attention to them.

## 4 EXPERIMENTS

### 4.1 EXPERIMENT SETUP

**Dataset.** We evaluate QFree-Det on two detection benchmark datasets: COCO2017 Lin et al. (2014) and WiderPerson Zhang et al. (2019). These two datasets differ in the number of training images and the variety of detection scenes. More dataset information is presented in Sec. B.1 of the **Appendix**. The ablation studies are performed on the COCO2017 dataset.

**Implementation details.** For fair comparison, we adopt the same training recipe from DINO (Zhang et al., 2022b) and train models with the AdamW (Loshchilov & Hutter, 2017) optimizer. QFree-DINO utilizes *4-scale* features from the backbone. The models are trained with a mini-batch size 8 on Tesla V100 GPUs. In the ablations, our models are trained for 12 epochs (1× training scheduler).

**Evaluation criteria.** For COCO2017, we evaluate the detection performance using the standard average precision (AP) (Liu et al., 2021a) metric under various IoU thresholds and object scales, following the evaluation metrics in COCO (Lin et al., 2014). For WiderPerson, we employ the evaluation metrics of AP, Recall, and mMR, commonly used in pedestrian detection (Zhang et al., 2019; Rukhovich et al., 2021).

## 4.2 MAIN RESULTS

**Results on COCO2017.** Table 3 presents a comprehensive comparison of our QFree-Det with multiple popular detectors using *various backbones* across *different* training epochs. It can be observed that QFree-Det achieves overall the *best* performance across five different backbones He et al. (2016); Cao et al. (2023); Liu et al. (2021b; 2024) in metrics of AP and $AP_S$ for general object detection and small object detection, respectively. For the ResNet50 backbone, our model outperforms the baseline model DINO by **+1.5%** AP and **+2.3%** $AP_S$ under $1\times$ scheduler (12 epochs). In particular, QFree-Det (**24** epochs only) obtains higher performance by **+0.4%** AP (51.3% vs. 50.9%) and **+0.9%** $AP_S$ (35.5% vs. 34.6%) than DINO (36 epochs), clearly demonstrating its training efficacy and effectiveness. For the backbone of Strip-MLP-T and Swin-T (where relatively fewer methods have reported results), our models achieve new SOTA, **55.0%** AP and **54.9%** AP with parameters of 45M and 49M, respectively. When compared to the larger backbone of Swin-L Liu et al. (2021b), our model surpasses DINO with a notable improvement of **+0.9%** AP (57.7% vs. 56.8%).

Recently, the visual state space model Gu & Dao (2023) is introduced to address the quadratic complexity of the attention mechanism, and MambaOut Yu & Wang (2024) has pointed out that visual Mamba has great potential on long-sequence visual tasks like OD. Therefore, we test QFree-Det and DINO with the backbone of VMamba-T Liu et al. (2024). Table 3 shows that QFree-Det creates a new SOTA result of **54.4%** AP and **38.8%** $AP_S$ under $1\times$ training schedule (12 epochs), higher than DINO by **+0.9%** AP and **+2.2%** $AP_S$. Moreover, when compared to DiffusionDet Chen et al. (2023b), which is the only existing model capable of predicting a *free* number of objects, QFree-Det (24 epochs) significantly outperforms DiffusionDet (60 epochs) with an increase of **+4.5%** AP and **+5.9%** $AP_S$, clearly demonstrating its superiority.

**Results on WiderPerson.** To further evaluate the effectiveness of QFree-Det, we conduct another experiment on the challenging WiderPerson dataset. Following the recipe of previous works Zhang et al. (2019); Rukhovich et al. (2021), we present results on the "Hard" subset of annotations in Table 4. QFree-Det obtains higher performance across all metrics over the baseline model of DINO using four different backbones, and outperforms other advanced models, demonstrating its effectiveness once again.

Table 4: Results on WiderPerson. The symbol † means the model is trained by us using the official code.

| Method | Year | Epoch | AP↑ | Recall↑ | mMR↓ |
|---|---|---|---|---|---|
| PS-RCNN (Ge et al., 2020) | 2020 | 12 | 89.96 | 94.71 | - |
| IterDet-2-iter (Rukhovich et al., 2021) | 2021 | 24 | 91.95 | 97.15 | 40.78 |
| He et al. (He et al., 2022) | 2022 | - | 91.29 | - | 40.43 |
| Cascade Transformer (Ma et al., 2023) | 2023 | 50 | 92.98 | 97.66 | 38.41 |
| DINO-ResNet50† | 2022 | 24 | 92.75 | 99.08 | 40.08 |
| **QFree-Det-ResNet50 (ours)** | 2024 | 24 | **93.24** | **99.57** | **39.47** |
| DINO-Strip-MLP-T† | 2022 | 24 | 93.19 | 99.42 | 38.21 |
| **QFree-Det-Strip-MLP-T (ours)** | 2024 | 24 | **93.75** | **99.65** | **38.11** |
| DINO-Swin-T† | 2022 | 24 | 93.07 | 99.42 | 38.78 |
| **QFree-Det-Swin-T (ours)** | 2024 | 24 | **93.67** | **99.65** | **38.05** |
| DINO-VMamba-T† | 2024 | 24 | 93.43 | 99.36 | 38.76 |
| **QFree-Det-VMamba-T (ours)** | 2024 | 24 | **94.04** | **99.65** | **37.04** |

## 4.3 ABLATION STUDIES

**Ablation on the components of QFree-Det.** We evaluate the impact of different components in QFree-Det using DINO-ResNet50 Zhang et al. (2022b) as the basic model. Our baseline model undergoes a preliminary transformation from fixed-number object detection to free-object detection through dynamic selection from encoder tokens and the removal of the SA mechanism. The results in Table 5 highlight the effectiveness of the proposed AFQS algorithm, PoCoo loss, and the LDD framework in improving the model's performance.

Table 5: Ablation on the components of QFree-Det. The *free-c* means "free-conditioned query", indicating the query is still constrained by the testing parameters.

| AFQS | PoCoo | LDD | QNum | AP | $AP_{50}$ | $AP_{75}$ | $AP_S$ | $AP_M$ | $AP_L$ |
|---|---|---|---|---|---|---|---|---|---|
| | | | fixed | 49.0 | 66.6 | 53.5 | 32.0 | 52.3 | 63.0 |
| | | | *free-c* | 44.1 | 59.5 | 48.1 | 27.9 | 47.6 | 56.7 |
| ✓ | | | free | 48.7 (+4.6) | 65.8 | 53.4 | 31.6 | 52.1 | 62.4 |
| ✓ | ✓ | | free | 49.3 (+0.6) | 65.6 | 53.9 | 32.2 | 52.6 | 63.4 |
| ✓ | ✓ | ✓ | free | **50.5 (+1.2)** | **67.5** | **55.1** | **34.3** | **54.6** | **64.5** |

**Ablation on the number of SA in DP.** SA plays a crucial role in removing duplicate detections. We conduct ablations on the number of SA to assess the impact of SA in DP and determine the optimal configuration. Table 6 demonstrates that the absence of SA noticeably decreases model performance, dropping to 34.7% AP. This significantly highlights the effectiveness and necessity of

SA in the decoder of transformer-based detectors. To balance computational efficiency and accuracy, we utilize 2 layers of SA ($\lambda = 2$) in DP for other experiments. Notably, the first four BLP layers for object localization do not incorporate SA, while only the last two DP layers each use two SA (**4 in total**) for deduplication. This design remains efficient, achieving comparable or lower computational costs than DINO, which employs **6** SA modules.

Table 6: Ablation on the number of SA.

| $\lambda$ | AP | $AP_{50}$ | $AP_{75}$ | $AP_S$ | $AP_M$ | $AP_L$ |
|---|---|---|---|---|---|---|
| 0 | 34.7 | 46.1 | 38.0 | 24.4 | 39.3 | 44.7 |
| 1 | 50.1 | 67.1 | 54.7 | 33.7 | 53.8 | 64.4 |
| 2 | **50.5** | **67.5** | **55.1** | **34.3** | **54.6** | 64.5 |
| 3 | 50.4 | 67.5 | 55.1 | 33.6 | 54.1 | **65.0** |

**Ablation on PoCoo loss.** To evaluate the effectiveness of PoCoo loss, we conduct ablations and compare it to the BCE loss and IA-BCE Cai et al. (2023) loss. The results in Table 7 demonstrate the significant effectiveness of our PoCoo loss. It outperforms BCE loss and IA-BCE loss

Table 7: Ablations on different loss functions.

| Loss Type | AP | $AP_{50}$ | $AP_{75}$ | $AP_S$ | $AP_M$ | $AP_L$ | AR | $AR_S$ |
|---|---|---|---|---|---|---|---|---|
| BCE | 49.0 | 67.3 | 53.4 | 32.6 | 52.4 | 63.0 | 74.3 | 59.6 |
| IA-BCE | 50.1 | 66.8 | 54.7 | 33.1 | 53.9 | **65.2** | 74.1 | 59.0 |
| PoCoo | **50.5** | **67.5** | **55.1** | **34.3 (+1.2)** | **54.6** | 64.5 | **74.4** | **60.0** |

by **+1.7%** and **+1.2%** in terms of $AP_S$, respectively. In addition, it can be observed that PoCoo loss also achieves higher performance in average recall for small objects ($AR_S$) than other losses, with a increase of **+0.4%** and **+1.0%** over BCE loss and IA-BCE loss for small objects, respectively. The consistently higher average precision and recall metrics for the PoCoo loss clearly demonstrate its effectiveness in accurately detecting more small objects, compared to other loss functions.

**Experiment on other transformer-based detector.** To further show the effectiveness of our QFree-Det on other transformer-based detectors beyond DINO, we further conducted another experiment on the other transformer-based detector like Deformable DETR Zhu et al. (2020). We developed a new variant of Deformable DETR by applying our AFQS, PoCoo loss, and the LDD to Deformable DETR model, denoted as Deformable DETR-QFree (DD-QFree in Table 8). The results presented in Table 8 show that our model outperforms the original Deformable DETR by **+3.1%** AP and **+3.5%** $AP_S$ and converges faster (in Fig. 3) on COCO dataset, further demonstrating the effectiveness and generalizability of our approach.

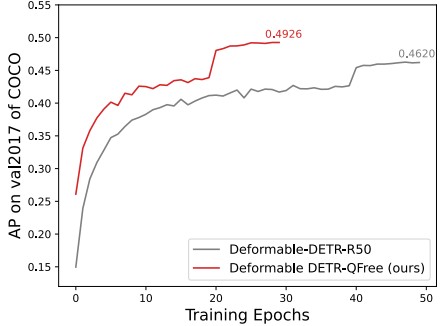

Figure 3: Convergence curves of Deformable DETR and Deformable DETR-QFree model.

**Other experiments and ablations in the Appendix.** Due to the limited space, more experiments and ablations are presented in Sec. B of **Appendix**, including experimental tests about: (1) the proportional relation between the number of objects and the number of queries; (2) performance on challenging dense objects

Table 8: The experiment results about the Deformable DETR and its variants with our approaches on val2017 of COCO dataset.

| Method | Epochs | AP | $AP_{50}$ | $AP_{75}$ | $AP_S$ | $AP_M$ | $AP_L$ |
|---|---|---|---|---|---|---|---|
| Deformable DETR | 50 | 46.2 | 65.2 | 50.0 | 28.8 | 49.2 | 61.7 |
| **DD-QFree (ours)** | **30** | **49.3 (+3.1)** | **66.4** | **53.6** | **32.3 (+3.5)** | **52.7** | **63.9** |

of WiderPerson dataset; and ablations on the (1) classification threshold $S$ in AFQS; (2) architecture configuration of LDD; (3) positional query; (4) connection order of CA and SA in DP; (5) one-to-many matching of $K$; (6) SGQ; (7) cost weight and (8) loss weight, respectively.

## 5 CONCLUSION

This paper proposes QFree-Det, a novel query-free detector capable of dynamically detecting a variable number of objects in different input images. QFree-Det addresses the limitation of "free-object predictions" by introducing the AFQS algorithm. For the "detecting ambiguity" issue, by rethinking the roles of SA and CA in the decoder, we design a novel LDD framework to decompose the detection process into two simple steps: box locating and object deduplication, with the sequential matching in our BLP and DP parts. Extensive experiments on diverse datasets demonstrate the effectiveness of QFree-Det across various backbone models. We hope that QFree-Det inspires the development of high-quality object detectors and multi-modal models in future research.

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

## A APPENDIX FOR METHOD

### A.1 CONTENT QUERY AND POSITIONAL QUERY

Based on DETR Carion et al. (2020), many follow-up works Zhu et al. (2020); Meng et al. (2021); Yao et al. (2021); Wang et al. (2022); Liu et al. (2022); Zhang et al. (2022b) have made efforts to represent the learned object queries in DETR more explicitly. These works propose different formulations and interpretations for the object queries. For instance, Conditional DETR Meng et al. (2021) and Anchor DETR Wang et al. (2022) formulate queries as learnable 2D coordinates $(x, y)$, which provide explicit spatial information for the cross-attention module in the transformer decoder. DAB-DETR Liu et al. (2022) reformulates the query with 4D box coordinates $(x, y, w, h)$ with better spatial priors. DAB-DETR constructs each query with two types: *content query* and *positional query*. The content query is initialized as *static embeddings*, similar to the *decoder embeddings* in DETR Carion et al. (2020). The positional query incorporates the position and size of each bounding box into the transformer decoder, enabling the measurement of query-to-feature similarity in the cross-attention module between the encoder features and the queries. Recently, advanced works, such as DINO Zhang et al. (2022b) and Stable-DINO Liu et al. (2023a), have followed these two types of queries and achieved promising performance.

### A.2 HOW DOES THE SA MODULE REDUCE DUPLICATE DETECTIONS?

In object detection, deduplicating detected objects is extremely challenging Carion et al. (2020); Cheng et al. (2023). Commonly, NMS is used as a post-process to remove duplicate bounding boxes based on overlap (IoU). However, this approach relies on manual thresholds and can mistakenly remove overlapping objects, hurting performance. In transformer-based detectors, where each query predicts only one object, the similarity between queries can be used to deduplicate the predictions, enabling end-to-end training. SA computes query similarities via attention scores, which are obtained by computing attention maps between all *pairwise* queries. For these similar queries, with the one-to-one matching label alignment mechanism, each query matched to a ground truth box will gradually obtain a higher classification score under the supervision of the loss function, while unmatched queries will gradually obtain lower scores. Ultimately, by reducing classification scores of similar queries, SA can effectively deduplicate the predictions.

### A.3 ALIGNMENT OF QUERY NUMBER IN THE SAME BATCH OF AFQS

In Sec. 3.3, we present our AFQS algorithm, which shifts the fixed number of queries into dynamic initialization and achieves "free-object predictions". Ensuring that the same number of queries is used for different images within the same batch is necessary to facilitate effective training during the training stage. For the training batch size $b$, we can get the number of queries from each image within the batch: $N_{query} = \{n_1, ..., n_b\}$. Then, we determine the *maximum* value among $N_{query}$ as the batch query number $N_{query}^b$, which ensures that a sufficient number of queries are selected for all images. Notably, additional **placeholder queries** must be selected for images where the number of queries filtered by the AFQS algorithm is less than $N_{query}^b$. Finally, we sort the classification scores of encoder tokens in ascending order and choose the tokens with low scores as placeholder queries to minimize the similarity between the placeholder queries and non-placeholder queries.

### A.4 DIFFERENCE WITH EXISTING ONE-TO-MANY METHODS

One-to-many matching label assignment is a common and significant approach Hu et al. (2024) to accelerate the model convergence and enhancing the training efficiency. The proposed QFree-Det model is fundamentally different from existing one-to-many matching approaches, such as Hybrid-Matching Jia et al. (2023) and DAC-DETR Hu et al. (2024), which simply mixes the one-to-one and one-to-many matching by the auxiliary decoder branches with shared weights. The differences are in the following ways: **(1) Motivation**: QFree-Det specifically aims to address the issue of *detecting ambiguity* that arises when combining one-to-one and one-to-many matching approaches. It serves as an effective solution to address this issue for other transformer-based detectors. **(2) Implementation**: by decomposing the detection process into two simle steps: boxes locating with BLP module and objects deduplication with DP module, we construct a novel decoder of LDD that utilizes se-

quential matching to alleviate matching ambiguity. In contrast, other methods achieve one-to-many matching through additional decoder branches, which increase the complexity and training cost of the model. **(3) Performance**: as shown in Table 3, QFree-Det outperforms other one-to-many matching methods, such as $\mathcal{H}$-Deformable-DETR and DAC-DETR. Additionally, our QFree-Det significantly surpasses another free-form model of the diffusion-based Diffusion-Det Chen et al. (2023b) model by **+4.5%** AP (51.3% in 24 epochs of QFree-Det vs. 46.8% in 60 epochs of Diffusion-Det, Table 3), further confirming its effectiveness.

### A.5 COMPLEXITY ANALYSIS

QFree-Det is a novel query-free detector that can adaptively select a variable number of queries with the different input images, as shown in Fig. 9, Fig. 10 and Fig. 11. Due to the dynamic computational complexity resulting from the adaptive query selection process, we use a fixed 900 queries for QFree-Det to calculate the FLOPs in Table 3 of Sec. 4.2 in the main text for fair comparison with other models.

Actually, the classification threshold $S$ affects the number of selected queries: a higher $S$ leads to fewer queries, and vice versa. Taking the QFree-Det-ResNet50 model with a pool size $P$ of 900 as an example, our model can effectively reduce the number of queries, and the corresponding FLOPs are also reduced, as shown in Fig. 4. The ablation results on the $S$ are presented in Table 11 of Sec. B.3.

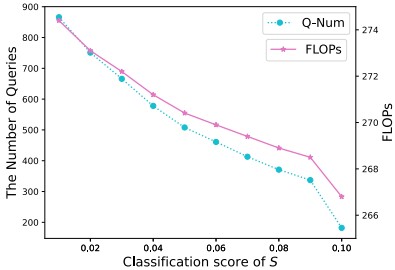

Figure 4: The number of queries and FLOPs of the model w.r.t threshold of classification score $S$.

## B APPENDIX FOR EXPERIMENT

### B.1 DETAILS ABOUT DATASETS

**COCO2017.** The COCO (Lin et al., 2014) dataset is a widely used benchmark dataset for object detection. COCO2017 consists of 118k training images and 5k validation images, with over 80 object categories.

**WiderPerson.** WiderPerson (Zhang et al., 2019) is a large and diverse dataset for dense pedestrian detection in real-world settings. It consists of 13,382 images with a total number of 399,786 annotations, averaging 29.87 annotations per image. This dataset presents significant challenges for SOD due to its diverse scenarios and substantial occlusion. It includes 8,000 images for training and 1,000 images for validation.

### B.2 EXPERIMENTAL TEST

**The proportional relation between the number of objects and the number of queries.** It is intuitive that the more potential objects to be detected, the more queries would be required. To verify the effectiveness of AFQS in adpatively query selection, we conducted an additional test to observe the trends in model accuracy and the dynamic selection of query quantity as the number of objects in the test images increases. Specifically, we divided the COCO validation set into 10 subsets based on the number of objects in each image, with a step size of 5 objects per image. Then, we tested the performance and counted the number of queries selected by the QFree-Det model for each subset. As shown in Table 9, the number of queries selected by the model increases as the number of objects to be detected increases, and the growth rate gradually becomes slow, as illustrated in the Fig. 5. At the same

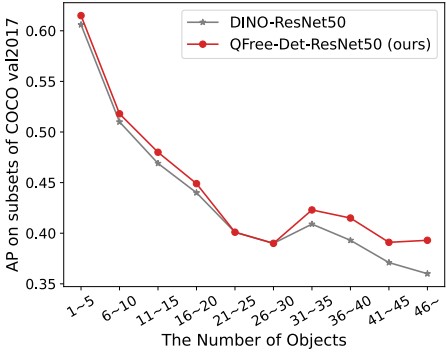

Figure 5: The performance comparison on the subsets of val2017 of COCO. The subsets are generated based on the number of objects per image with a step size of 5.

Table 9: The performance (AP) on the subsets of val2017 of COCO.

| Object Numbers | 1-5 | 6-10 | 11-15 | 16-20 | 21-25 | 26-30 | 31-35 | 36-40 | 41-45 | 46+ |
|---|---|---|---|---|---|---|---|---|---|---|
| Average Objects | 2.60 | 7.70 | 13.02 | 17.69 | 22.74 | 27.35 | 33.31 | 37.68 | 42.77 | 55.00 |
| Image Numbers | 2769 | 985 | 556 | 324 | 170 | 83 | 29 | 22 | 9 | 5 |
| Query Numbers | 65.22 | 180.46 | 312.47 | 443.98 | 537.91 | 645.16 | 699.82 | 812.41 | 872.00 | 900.00 |
| Query Num/Object Num | 25.08 | 23.44 | 23.99 | 25.10 | 23.65 | 23.59 | 21.01 | 21.56 | 20.39 | 16.37 |
| DINO | 60.6 | 51.0 | 46.9 | 44.0 | 40.1 | 39.0 | 40.9 | 39.3 | 37.1 | 36.0 |
| **QFree-Det (ours)** | **61.5** | **51.8** | **48.0** | **44.9** | **40.1** | **39.0** | **42.3** | **41.5** | **39.1** | **39.3** |

time, our model obtained overall higher performance across all subsets, further validating its effectiveness, as presented in Table 9 and Fig. 5.

When processing images with more objects, the advantages of our method become more apparent, outperforming DINO by +2.0% AP and +3.3% AP in the subsets of 41-45 and over 46, respectively. The trend of the query number starts to slow down as the number of objects in the image increases, as shown in Fig. 6. The subset of 1-5 occupies 55.9% images among val2017 of COCO. For this subset, our model only uses **7.25%** of the queries (65.22 vs 900) while achieving higher performance by **+0.9% AP**, indicating that the selected queries via AFQS are more effective. This demonstrates a significant advantage in common scenarios, as it can effectively reduce computational costs. Moreover, the number of queries selected by our model can be adaptively adjusted based on

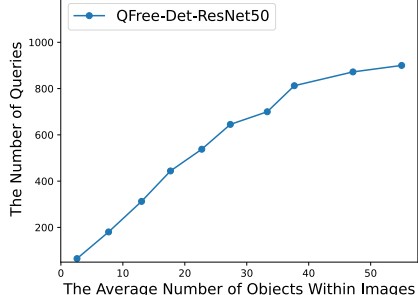

Figure 6: The relation between the number of objects and queries dynamically selected by the QFree-Det.

different classification thresholds , without the need to retrain the model, making it adaptable to different detection scenarios more easily.

**Performance on challenging dense objects detection of WiderPerson dataset.** WiderPerson is a large, diverse and challenging dataset for dense pedestrian detection, with an average of 29.87 annotations per image. As illustrated in Fig. 7, the statistic results on its validation set indicate that there are 679 images with less than 30 objects, and 321 images with 30 or more objects. To evaluate the effectiveness of our method in handling the more challenging scenario with dense objects over 30 of the WiderPerson dataset, we divided this validation set into two test subsets. The results in Table 10 demonstrate that, across four backbone models, our models consistently achieve overall higher AP, Recall, and lower mMR, further validating its effectiveness.

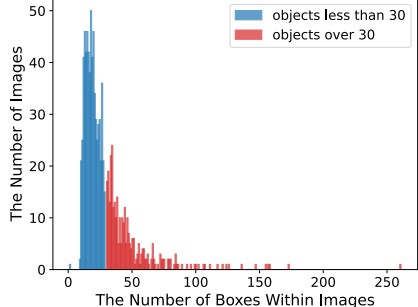

Figure 7: The histogram of the WiderPerson validation dataset.

Table 10: The experiment results on the WiderPerson dataset.

| Objects Per Image | Less Than 30 (679 images) | | | Over 30 (321 images) | | |
|---|---|---|---|---|---|---|
| Method | AP↑ | Recall↑ | mMR↓ | AP↑ | Recall↑ | mMR↓ |
| DINO-ResNet50 | 95.84 | 99.63 | **26.71** | 88.84 | 98.45 | 57.18 |
| **QFree-Det-ResNet50 (ours)** | **96.10** | **99.80** | 27.50 | **89.65** | **99.31** | **55.09** |
| DINO-Strip-MLP-T | 96.27 | 99.80 | 26.16 | 90.07 | 99.40 | 56.33 |
| **QFree-Det-Strip-MLP-T (ours)** | **96.39** | **99.83** | **26.03** | **90.45** | **99.45** | **53.37** |
| DINO-Swin-T | 96.45 | 99.71 | **26.35** | 90.06 | 99.00 | 55.58 |
| **QFree-Det-Swin-T (ours)** | **96.52** | **99.84** | 26.36 | **90.08** | **99.43** | **53.63** |
| DINO-VMamba-T | 96.09 | **99.83** | 27.12 | 90.09 | 99.46 | 53.68 |
| **QFree-Det-VMamba-T (ours)** | **96.60** | 99.80 | **25.68** | **90.76** | **99.47** | **52.08** |

### B.3 ABLATION STUDIES

**Ablation on the classification threshold $S$ in AFQS.** The classification threshold $S$ in AFQS enables adaptive control of the number of decoder queries. A higher value of $S$ reduces the number of decoder queries, while a lower $S$ increases them, as illustrated in Fig. 4. We conduct ablation experiments with varying $S$ to evaluate its impact on the model performance, as shown in Table 11. The smaller $S$ brings higher performance, and when $S$ ranges from 0.01 to 0.05, the model performance varies in a small range of $50.2\% \sim 50.6\%$ AP, indicating good robustness. Notably, when $S = 0.10$, our model used only

Table 11: Ablation results of threshold of $S$ in AFQS.

| $S$ | AP | $AP_{50}$ | $AP_{75}$ | $AP_S$ | $AP_M$ | $AP_L$ |
|---|---|---|---|---|---|---|
| 0.01 | **50.6** | **67.9** | **55.4** | 34.1 | 54.3 | 64.8 |
| 0.02 | 50.5 | 67.5 | 55.1 | **34.3** | **54.6** | 64.5 |
| 0.03 | 50.2 | 67.4 | 54.6 | 33.2 | 53.7 | 64.8 |
| 0.04 | 50.2 | 67.2 | 54.5 | 33.6 | 53.8 | 64.5 |
| 0.05 | 50.3 | 67.5 | 54.8 | 33.4 | 54.0 | **65.0** |
| 0.06 | 50.1 | 67.3 | 54.6 | 33.2 | 53.7 | 64.5 |
| 0.07 | 49.8 | 66.7 | 54.5 | 32.8 | 53.4 | 64.1 |
| 0.08 | 50.0 | 67.0 | 54.5 | 33.2 | 53.9 | 64.0 |
| 0.09 | 49.6 | 66.7 | 54.0 | 32.1 | 53.3 | 63.8 |
| 0.10 | 49.5 | 66.4 | 54.0 | 33.2 | 53.0 | 64.0 |

20.2% of the DINO queries on average, yet achieved a **+0.5%** higher AP than DINO (49.5% vs. 49.0%). These results strongly demonstrate the significant effectiveness of our AFQS approach for "free-object predictions".

**The architecture configuration of LDD.** To maintain consistency with methods such as DETR and DINO, we design LDD architecture with six layers, ensuring that the parameter count remains unchanged. However, the distribution of BLP and DP within these six layers significantly affects the model's performance: excessive BLP layers can hinder the model's ability to eliminate duplicate detections effectively; conversely, an excessive number of

Table 12: Ablation on the BLP and DP layers.

| BLP | DP | AP | $AP_{50}$ | $AP_{75}$ | $AP_S$ | $AP_M$ | $AP_L$ |
|---|---|---|---|---|---|---|---|
| 1 | 5 | 49.5 | 67.1 | 54.2 | 33.3 | 53.1 | 63.5 |
| 2 | 4 | 49.4 | 67.2 | 53.8 | 33.4 | 52.9 | 63.3 |
| 3 | 3 | 50.0 | 67.1 | 54.6 | 33.3 | 53.7 | 64.5 |
| **4** | **2** | **50.5** | **67.5** | **55.1** | **34.3** | **54.6** | 64.5 |
| 5 | 1 | 50.0 | 66.7 | 54.3 | 32.9 | 53.8 | **64.7** |

DP layers may lead to inaccurate bounding box predictions and decreased training efficiency. To determine the optimal configuration, we perform ablations on the number of BLP and DP layers in LDD. As presented in Table 12, the best performance across all metrics is achieved with 4 BLP and 2 DP layers. We adopt this configuration for other experiments.

**Ablation on positional query.** In Sec. 3.1.2 of the main paper and Sec. A.1, we highlighted the limitation of a fixed number of queries in existing transformer-based detectors. To address this limitation, we introduced the new AFQS algorithm, which replaced the CQ and eliminated the need for PQ. To investigate the impact of positional queries in our QFree-Det model, we conduct ablation experiments by adding positional

Table 13: Ablation on the positional query in LDD.

| PQ in SA | PQ in CA | AP | $AP_{50}$ | $AP_{75}$ | $AP_S$ | $AP_M$ | $AP_L$ |
|---|---|---|---|---|---|---|---|
| ✓ | | 50.0 | 67.1 | 54.5 | 33.3 | 53.1 | 65.3 |
| | ✓ | 50.3 | 67.4 | 54.8 | 33.5 | 53.7 | 64.5 |
| ✓ | ✓ | 50.1 | 67.1 | 54.5 | 33.5 | 53.9 | **64.7** |
| | | **50.5** | **67.5** | **55.1** | **34.3** | **54.6** | 64.5 |

query to both the CA and SA modules in LDD. The results presented in Table 13 reveal that the inclusion of PQ leads to a certain degree of accuracy degradation, supporting the validity of our analysis. Furthermore, our proposed AFQS and SADQ methods simplify the model structure and reduce model complexity, compared to the one-to-many model with additional decoder branches.

**Ablation on the connection order of CA and SA in DP.** In existing transformer-based detectors, it is commonly observed that SA is connected before CA in the decoder, following the original architecture in DETR. However, this study argues that this connection scheme may introduce detecting ambiguity due to the opposing impacts of SA and CA on the object queries. We conduct an ablation study on QFree-Det

Table 14: Ablation on the order of the SA and CA.

| Connection | AP | $AP_{50}$ | $AP_{75}$ | $AP_S$ | $AP_M$ | $AP_L$ |
|---|---|---|---|---|---|---|
| SA → CA | 50.1 | 67.1 | 54.6 | 33.5 | 53.3 | **64.7** |
| **CA → SA** | **50.5** | **67.5** | **55.1** | **34.3** | **54.6** | 64.5 |

to investigate the effect of reversing the order of CA and SA connections. Table 14 demonstrates the effectiveness of our connection scheme of SA in DP, significantly enhancing the model's ability to remove duplicate detections.

**Ablation on the one-to-many matching of K.** One-to-many matching is the significant approach to enhance training efficiency by increasing the number of positive samples. We perform ablations on the ground truth box repeating times $K$ to determine the optimal configuration.

As K increases, the difficulty of removing duplicate box detections also increases. Conversely, smaller K values result in an insufficient number of positive samples, leading to decreased training efficiency. Table 15 shows the best performance is obtained with $K = 6$, which is used for other experiments.

Table 15: Ablation on K of one-to-many matching.

| $K$ | AP | $AP_{50}$ | $AP_{75}$ | $AP_S$ | $AP_M$ | $AP_L$ |
|---|---|---|---|---|---|---|
| 1 | 50.1 | 67.0 | 54.6 | 33.4 | 53.7 | 64.9 |
| 3 | 50.2 | 67.2 | 54.5 | 33.4 | 54.1 | 64.7 |
| **6** | **50.5** | **67.5** | **55.1** | **34.3** | **54.6** | 64.5 |
| 9 | 50.1 | 66.8 | 54.7 | 33.1 | 53.9 | **65.2** |
| 12 | 49.8 | 67.0 | 54.1 | 32.6 | 53.6 | 64.4 |

**Ablation on the stop gradient of queries (SGQ).** The SGQ plays a crucial role in separating the gradient flow of queries between the one-to-many matching of BLP and the one-to-one matching of DP. We conduct an ablation on SGQ to show its impact on performance. In Table 16, we observe that the absence of stop gradient of queries from DP to BLP leads to a 2.0% decrease in performance, emphasizing the necessity and effectiveness of our SGQ method for sequential matching to address the issue of detecting ambiguity.

**Ablation on the classification cost weight for the matching process.** Accurately matching predicted boxes with ground truth boxes is critical for transformer-based models. We employ the same cost components as DETR and DINO, including the L1 cost for bounding boxes, binary cross-entropy (BCE) cost for classification, and generalized Intersection over Union (GIoU) cost. The weights assigned to each cost component play a signifi-

Table 16: Ablation on SGQ.

| SGQ | AP | $AP_{50}$ | $AP_{75}$ | $AP_S$ | $AP_M$ | $AP_L$ |
|---|---|---|---|---|---|---|
| | 48.5 | 66.5 | 52.9 | 32.2 | 52.1 | 62.5 |
| ✓ | **50.5** | **67.5** | **55.1** | **34.3** | **54.6** | **64.5** |

cant role in optimizing the training process and affecting the model's performance. In our QFree-Det model, sequential matching primarily addresses the issue of duplicate detections through one-to-one matching classification supervision. To investigate the impact of different *classification costs* on model training, we conduct ablation experiments to determine the optimal configurations. The results in Table 17 show that the absence of the classification cost ($BCE_{BLP} = 0.0$) hinders the model's performance. Including the classification cost in the matching process can introduce semantic information, leading to more appropriate query matches. However, assigning a higher weight to the classification cost makes it more challenging for the model to predict bounding boxes accurately, as the query with higher classification score would be matched with the ground truth box rather than the query with a higher Iou. Based on the experimental results, we adopt a weight of 0.2 for BCE cost as our training parameter.

Table 17: Ablation on the classification cost weight in BLP.

| $BCE_{BLP}$ | $L1_{BLP}$ | $GIou_{BLP}$ | $BCE_{DP}$ | $L1_{DP}$ | $GIou_{DP}$ | AP | $AP_{50}$ | $AP_{75}$ | $AP_S$ | $AP_M$ | $AP_L$ |
|---|---|---|---|---|---|---|---|---|---|---|---|
| 0.0 | 5.0 | 2.0 | 2.0 | 2.0 | 2.0 | 48.9 | 66.0 | 52.9 | 31.3 | 52.6 | 63.3 |
| 0.1 | 5.0 | 2.0 | 2.0 | 2.0 | 2.0 | 50.0 | 67.0 | 54.6 | 33.4 | 53.4 | 64.8 |
| **0.2** | 5.0 | 2.0 | 2.0 | 2.0 | 2.0 | **50.5** | 67.5 | **55.1** | **34.3** | **54.6** | 64.5 |
| 0.3 | 5.0 | 2.0 | 2.0 | 2.0 | 2.0 | 49.9 | 67.0 | 54.3 | 33.3 | 53.5 | 64.4 |
| 0.5 | 5.0 | 2.0 | 2.0 | 2.0 | 2.0 | 50.2 | **67.7** | 54.7 | 34.0 | 53.7 | **64.8** |
| 0.8 | 5.0 | 2.0 | 2.0 | 2.0 | 2.0 | 49.4 | 66.9 | 53.7 | 33.1 | 52.9 | 64.4 |

**Ablation on the loss weight for BLP and DP.** In transformer-based detectors Carion et al. (2020), there are three main loss functions: BCE loss for classification, L1 and GIoU loss for bounding box regression. To investigate the impact of different loss weights on the encoder, BLP, and DP components of the model, we conduct ablation experiments using various weight values. As presented in Table 18, the baseline model (index 0) is achieved with the same weight as DINO. To improve the one-to-one classification accuracy of the model, we reduce the weight of the L1 loss in the DP component (index 1) and increase the weight of the classification loss (index 2). Building upon the baseline, we further increase the overall weight of the classification loss across the encoder, BLP and DP (index 3). We then test the effect of increasing the weights of both the classification loss and the GIoU loss (index 4, 5, and 6, respectively). Experiments of index 3 and 6 indicate that a higher weight on the L1 loss of bounding boxes is important for box refinement in DP component. Finally, we adopt the weight configuration of index 3 as the loss weights for other model training.

Table 18: Ablation on different loss weights for encoder, BLP and DP.

| Index | PoCoo$_{enc}$ | PoCoo$_{BLP}$ | L1$_{BLP}$ | GIou$_{BLP}$ | PoCoo$_{DP}$ | L1$_{DP}$ | GIou$_{DP}$ | AP | AP$_{50}$ | AP$_{75}$ | AP$_S$ | AP$_M$ | AP$_L$ |
|---|---|---|---|---|---|---|---|---|---|---|---|---|---|
| 0 (baseline) | 1.0 | 1.0 | 5.0 | 2.0 | 1.0 | 5.0 | 2.0 | 49.4 | 66.2 | 54.1 | 32.4 | 53.1 | 64.2 |
| 1 | 1.0 | 1.0 | 5.0 | 2.0 | 1.0 | 1.0 | 2.0 | 49.5 | 66.4 | 54.0 | 32.3 | 53.4 | 63.5 |
| 2 | 1.0 | 1.0 | 5.0 | 2.0 | 2.0 | 1.0 | 2.0 | 48.1 | 64.8 | 52.3 | 31.5 | 52.2 | 62.1 |
| **3** | **1.5** | **2.0** | **5.0** | **2.0** | **2.0** | **5.0** | **2.0** | **50.5** | **67.5** | **55.1** | **34.3** | **54.6** | 64.5 |
| 4 | 1.5 | 3.0 | 5.0 | 2.0 | 3.0 | 1.0 | 2.0 | 49.8 | 67.1 | 54.2 | 33.1 | 53.6 | 64.2 |
| 5 | 1.5 | 3.0 | 5.0 | 3.0 | 3.0 | 2.0 | 3.0 | 49.9 | 67.0 | 54.2 | 32.8 | 53.8 | 64.4 |
| 6 | 1.5 | 2.0 | 5.0 | 3.0 | 2.0 | 5.0 | 3.0 | 50.4 | 67.4 | 55.0 | 33.7 | 54.1 | **64.6** |

**Experiments on the CrowdHuman dataset.** To further demonstrate the effectiveness of our model, we conducted a new experiment on CrowdHuman Shao et al. (2018) dataset, which is also a challenging datasets for dense pedestrian detection in the wild. The results listed in Table 19 show that QFree-Det obtains overall higher performance compared to DINO variants, further confirming the effectiveness of our approach.

Table 19: The experiment results on the CrowdHuman dataset with full-body annotations.

| Method | Epochs | AP↑ | Recall↑ | mMR↓ |
|---|---|---|---|---|
| DINO-ResNet50 | 24 | 86.61 | 95.11 | 52.85 |
| **QFree-Det-ResNet50 (ours)** | 24 | **86.87** | **95.25** | **52.08** |
| DINO-Strip-MLP-T | 24 | **88.38** | 95.82 | 50.30 |
| **QFree-Det-Strip-MLP-T (ours)** | 24 | 87.92 | **95.99** | **49.83** |
| DINO-Swin-T | 24 | 87.71 | 95.71 | 51.81 |
| **QFree-Det-Swin-T (ours)** | 24 | **88.14** | **95.86** | **51.12** |
| DINO-VMamba-T | 24 | 87.44 | 95.45 | 51.54 |
| **QFree-Det-VMamba-T (ours)** | 24 | **88.46** | **96.20** | **50.28** |

## B.4 PERFORMANCE COMPARISON ON COCO

Figure 8 compares the performance of different transformer-based detectors on the standard detection benchmarks of the COCO dataset. The results indicate that our QFree-Det model has significant advantages in terms of training efficiency (AP-Epoch, AP$_S$-Epoch), fewer parameters (AP-Params, AP$_S$-Params) on the performance of general object detection and small object detection. Notably, QFree-Det dramatically enhances the detection capabilities of the baseline DINO model, effectively demonstrating the effectiveness of the QFree-Det approach.

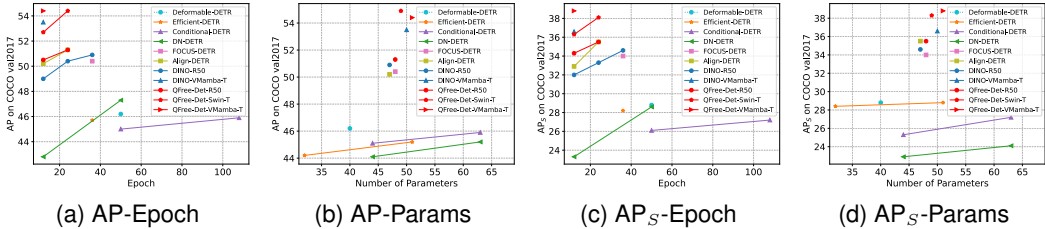

(a) AP-Epoch     (b) AP-Params     (c) AP$_S$-Epoch     (d) AP$_S$-Params

Figure 8: Comparison of transformer-based models of AP and AP$_S$ w.r.t the different number of parameters and training epochs on `val2017` of COCO.

## C VISUALIZATION OF FREE QUERIES ON `VAL2017` OF COCO

This paper proposes a novel transformer-based *query-free* detector that can predict a variable number of objects for different input images. QFree-Det effectively addresses the "fixed number of object predictions" limitation of transformer-based detectors. By adaptively selecting the number of queries from encoder tokens, our model significantly improves query efficiency by reducing redundant queries and decreasing the computational cost.

To visually demonstrate this, we compare the test results of our QFree-Det-ResNet50 (12 epochs) model and the DINO-ResNet50 (12 epochs) model on the `val2017` of COCO dataset in Fig. 9, Fig. 10, and Fig. 11. To ensure clear visualization, the queries are represented using solid circle points with a radius of 3, where the color of the circles indicates the different classification confidence scores. The detection bounding boxes are displayed using random colors to differentiate the different object instances.

Table 20: Inference speed tests on DINO and QFree-Det.

| Model | DINO | QFree-Det | QFree-Det | QFree-Det | QFree-Det | QFree-Det |
|---|---|---|---|---|---|---|
| Query Number | 900 | 1800 | 900 | 500 | 100 | 10 |
| Backbone (ms) | 15.2 | 15.2 | 15.2 | 15.2 | 15.2 | 15.2 |
| Encoder (ms) | 30.6 | 30.6 | 30.6 | 30.6 | 30.6 | 30.6 |
| Query Selection (ms) | 7.3 | 7.8 | 7.5 | 7.4 | 7.4 | 7.4 |
| Decoder (ms) | 14.1 | 11.0 | 9.2 | 8.9 | 8.9 | 8.8 |
| Inference Time (ms) | 67.2 | 64.6 | 62.5 | 62.1 | 62.1 | 62.0 |
| FPS (frame/s) | 14.9 | 15.5 | 16.0 | 16.1 | 16.1 | 16.1 |

In simple scenarios, such as the bear's detection in Fig. 9, QFree-Det-ResNet50 only used 2% of the queries compared to DINO-ResNet50, yet achieved accurate detection results. In the relatively complex scenarios with more objects in Fig. 10 and Fig. 11, QFree-Det-ResNet50 similarly adapted and selected fewer queries while achieving more precise detection results. This clearly demonstrates that the QFree-Det model can adaptively select the number of queries based on different image inputs, thereby enabling the "free-object predictions". This approach also reduces the number of redundant queries, effectively improving the model's performance and efficiency.

## D    INFERENCE SPEED TESTS AND ANALYSIS

In Sec. A.5 and Sec. B.2, we conducted experiments on COCO and WiderPerson, along with analyses to explore the relationship between the adaptive number of queries, computational complexity, and model accuracy. The results highlight the advantages of our method in both aspects.

To further show the effectiveness of the LDD architecture, we conduct additional tests on the inference speed of QFree-Det. For a fair comparison of our model with the baseline model DINO, the test was applied on the same codebase (official published code of DINO), backbone model (ResNet50), input image size ($1280 \times 800$), GPU device (RTX 3090), PyTorch lib (pytorch 2.2), and CUDA version (Driver Version: 535.183.01, CUDA Version: 12.2). We tested the DINO (900 query) model and QFreeDet (query varies from *10 to 1800*) model's inference speed, and the results are shown in the Table 20.

The inference time of both models is composed of four components: *backbone time, encoder time, query selection time, and decoder time*. Since the backbone and encoder components are identical in both models, their inference speeds are also the same. The inference time of query selection process in both models is similar, with most of the duration spent on the model classification head in predicting scores (about 7ms) among all encoder tokens for subsequent selection. For our AFQS algorithm, it then retrieves queries based on these scores within **1 ms**, which is faster in query selection to transform the fixed-query into a free-query for the DETR detector.

Additionally, we observed that under the same query conditions, such as using *900 queries*, our LDD framework significantly enhances the inference speed of decoder, achieving a **+34.8%** improvement compared to DINO. This result clearly validates the effectiveness of our designed novel LDD decoder framework. When the number of queries is further reduced, the decoder's inference speed remains stable, primarily due to the parallel computation performed by CA and SA in the decoder.

It is noted that to improve the model's inference speed is not the primary objective of our method. Table 20 indicates that the backbone and encoder models account for **73%** of the inference time. This observation provides insights for further optimizing these components to accelerate the inference speed of DETR models. Notably, our efficient LDD decoder framework has successfully increased the inference speed of the model's decoder by **34.8%**. Thus, this framework can be integrated with other model architectures, such as the YOLO series, not only to further enhance overall inference speed but also achieve the detection of a free number of objects.

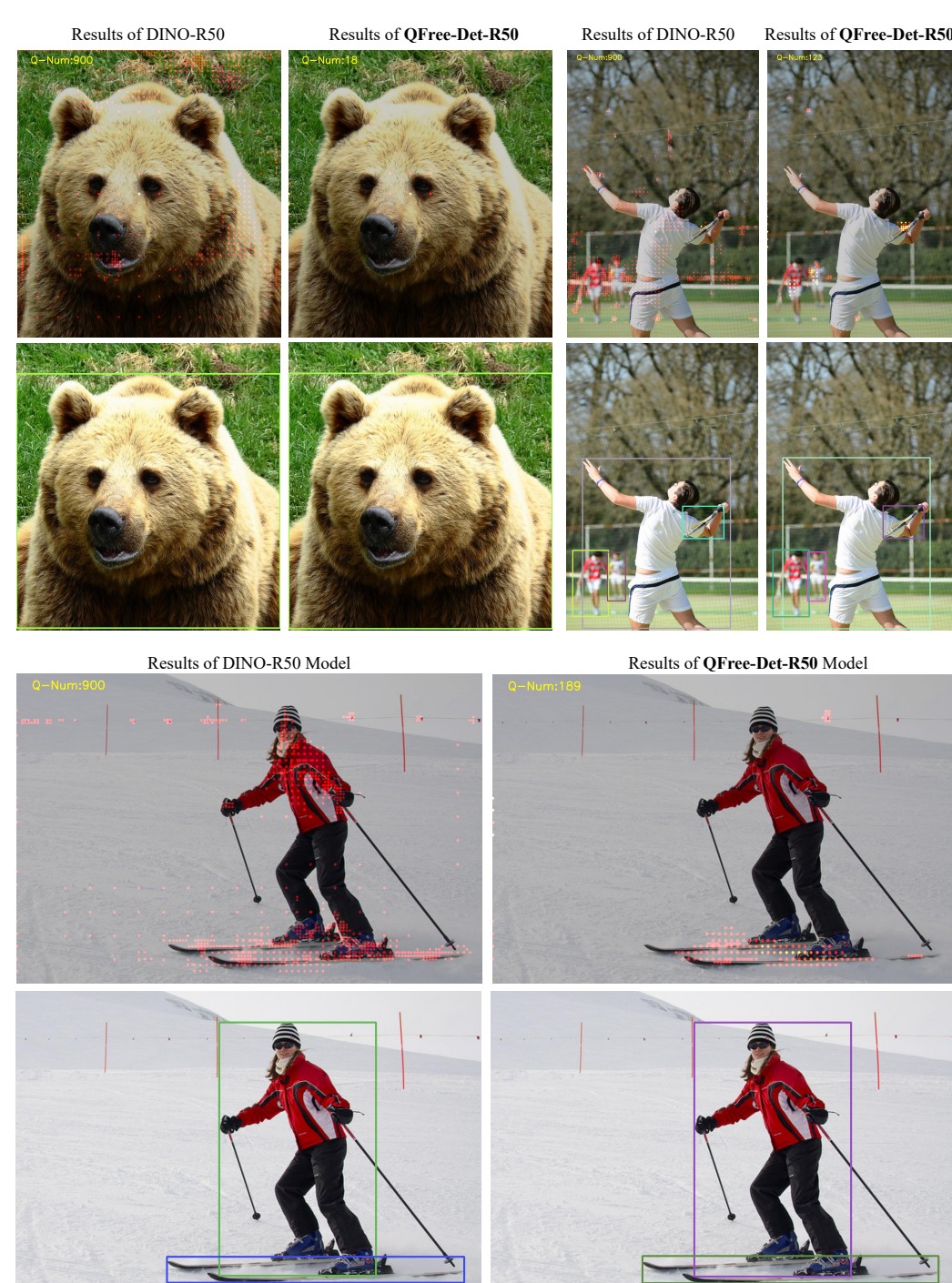

Figure 9: Visualization of queries in simple scenarios. The number of queries (Q-Num) selected for each image is on the top left corner of the corresponding image.

The primary limitation of *query-fixed* detectors is their requirement to predict a **large** and **fixed** number of detection results for both **sparse** and **dense** scenes. On one hand, this results in a considerable amount of *unnecessary and redundant computation*. On the other hand, when these fixed-query detectors are applied to more challenging downstream tasks, such as Open-Ended Detection Lin et al. (2024), these redundant queries would be further fed into the **large language model** (LLM) to gen-

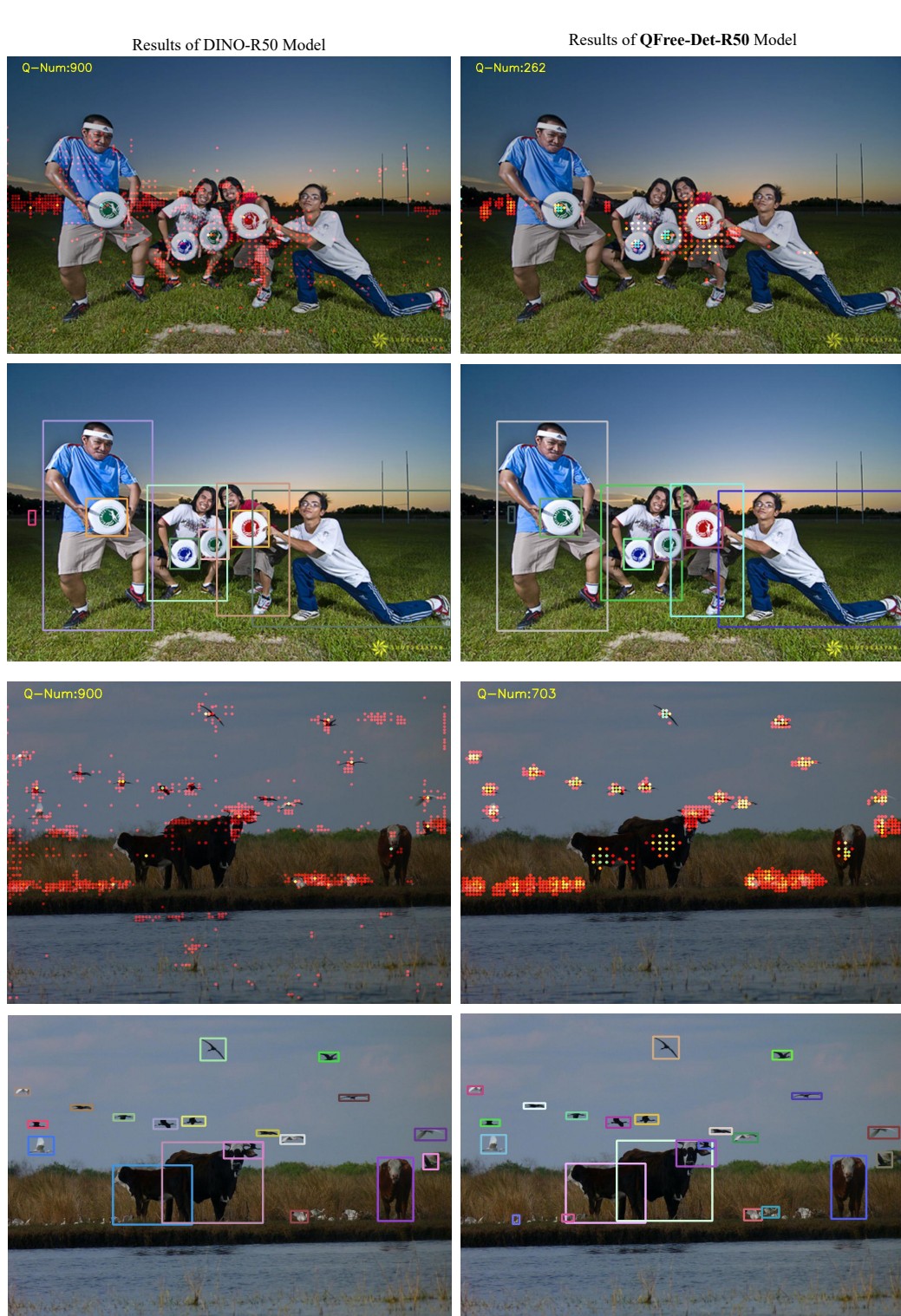

Figure 10: Visualization of queries in complex scenarios.

erate object names directly, which will significantly increase the overall computational complexity and cost.

Figure 11: Visualization of queries in complex scenarios.

To demonstrate this, we conducted a simple test using the public official code of GenerateU Lin et al. (2024). We evaluated the computational complexity of the generative large language model used in GenerateU. With 900 queries of the detector fed into the LLM, the complexity is **10,139.64**

**GFLOPs**. In contrast, with 200 queries, the complexity dropped to **2,253.45 GFLOPs**, which indicates that reducing the number of queries can significantly decrease the computational load of the LLM model for the open-ended detection multi-modal task.

Based on the analysis, our query-free model, which adapts the number of queries based on the image itself, *holds significant potential for open-ended object detection task*. This adaption characteristic enables reduced queries to feed into generative large language model, thereby decreasing the workload for subsequent category generation processing, which highlights its potential applications and importance for the future research in vision community.

# E  DISCUSSION

## E.1  THE DEDUPLICATION ROLES OF SELF-ATTENTION (SA) AND NMS

Removing duplicate detection boxes is a crucial step for reducing false positive samples in detection systems. We categorize existing methods of deduplication into two types based on their principles: **box-based** and **class-based**. Box-based methods, such as NMS, work by *comparing the overlapping (IoU) between predicted boxes*. While straightforward, they are sensitive to the IoU threshold and struggle with overlapping targets. In contrast, class-based methods utilize self-attention to compare features between queries, influencing classification scores to achieve deduplication, which results in *lower scores for redundant boxes*. This approach can mitigate the issue of overlapping objects and is commonly employed in DETR-like models. Intuitively, combining both *box-based* and *class-based* methods may further enhance the model's overall performance.

DETA Ouyang-Zhang et al. (2022) and DDQ-DETR Zhang et al. (2023) both use NMS to eliminate duplicates in their models. Compared to DETA and DDQ-DETR, our approach **addresses different problems** (in motivation and objective), **introduces different solutions to issues** (in query selection, decoder architecture, and loss function), and **achieves comparable results** in performance. In addition, *these differences do not diminish the contributions of each method to solving the respective issues.*

DETA is a *box-based* method for deduplication. Specifically, DETA is designed to investigate the impact of *one-to-many assignment-based training* on enhancing the training efficiency for DETR models, which **differs** to our motivation and objective. It achieves this by employing one-to-many **IoU assignments** in conjunction with NMS method, which is applied during both query selection and final prediction post-processing. This study shows that the *one-to-many IoU assignments*, *combined with NMS*, effectively improve training efficiency.

DDQ-DETR combines both *box-based* and *class-based* deduplication methods. It specifically explores how *query distinctness* affects the model's optimization process and accuracy, which also **differs** to our motivation and objective. DDQ-DETR introduces the Distinct Query Selection (DQS) module, which uses training-unaware NMS to filter dense queries into distinct queries (*box-based method*). Then, DDQ-DETR further applies Hungarian matching (*class-based method*), considering both bounding box scores and class scores, to generate final one-to-one detection results. Additionally, DDQ-DETR also uses one-to-many label assignments and incorporates an *auxiliary head* along with Auxiliary Loss for Dense Queries to maintain training efficiency (but this kind of mixing label assignments on same decoder weights still introduces the issue of "detection ambiguity"). Overall, this approach highlights that both sparse and dense queries in end-to-end detection are problematic. By explicitly combining box-based and class-based methods, DDQ-DETR ultimately enhances model accuracy.

Unlike these two methods, our approach focuses on transforming a *fixed-query* detector into a *free-query* detector, thereby addressing the fixed capacity of DETR-like models. We deeply explore the role of SA (which has NOT been examined in DDQ-DETR) to address challenges related to the existing decoder structure during this transition, particularly the training GPU memory demands issue associated with SA. In contrast to DETA and DDQ-DETR, *our object is not to investigate how to enhance model accuracy through NMS*. Furthermore, we observe that the existing connection between

CA and SA can result in the "**recurrent shifting**" problem (as noted in Sec. 3.1.3). To tackle these challenges, we developed a more effective decoder structure (LDD) and implemented *decoupled* one-to-one and one-to-many label assignments. This design significantly alleviates the "detection ambiguity" issue (Table 12 and Table 16). Notably, DETA and DDQ-DETR share a similar decoder structure with existing methods, which highlights the contribution of our approach of novel LDD decoder. Compared to previous class-based deduplication method, QFree-Det further *optimizes the decoder structure, enabling query-free detection while minimizing the "detection ambiguity" associated with one-to-many assignments*.

**Deduplication Part (DP) vs. NMS.**    DP is an integral part of the decoder and not a *post-processing* algorithm. Although we designed DP to eliminate duplicates, it is distinct from NMS and cannot be directly replaced by it. Our model's decoder, similar to those in existing DETR-like models, comprises six layers: 4 LBP layers and 2 DP layers, connected in series. As analyzed in DETR Carion et al. (2020), the decoder primarily reasons about the relations of queries and image context to generate detections, which is *necessary* for the detector. In Table 12, we conducted ablations on the BLP and DP layers. The results indicate that reducing the number of DP layers (for example, using only one DP layer) is harmful for the performance, underscoring the importance of the DP layer.

In Table 2, we present a *free-query* model that removes all SA (approximating the removal of the DP module), while employing the NMS to eliminate duplications. This model achieved only 47.5% AP on COCO, which is significantly lower than the 50.5% AP achieved with our model with DP, clearly highlighting the importance and irreplaceability of the DP.

On the other hand, NMS can only perform deduplication on the detection results that have already been obtained, and it is sensitive to parameters; it cannot generate detection results directly, making it unsuitable as a replacement for the decoder. The DETR model was originally designed to simplify the detection process in an **end-to-end** pipeline, removing post-processing steps like NMS and improving model robustness. *Our method aligns well with this goal*, enabling the model to function effectively without relying on NMS, but not to pull NMS back again.

The original DETR Carion et al. (2020) also notes that the FFN (Feed Forward Network, FFN) in the decoder has a significant impact on model accuracy. We have made a test on FFN of DINO model to reduce computational complexity by decreasing the hidden layer dimension in the FFN from 2048 to 768. However, this change resulted in a **1.6%** decrease in AP, underscoring the critical role of the decoder network layers in maintaining performance. This further underscores the importance of the decoder.

### E.2    THE DIFFERENCES OF SEQUENTIAL MATCHING WITH EXISTING METHODS

One-to-one matching and one-to-many matching label assignments have been demonstrated in several studies (such as H-DETR Jia et al. (2023), Align-DETR Cai et al. (2023), and DAC-DETR Hu et al. (2024), etc) to enhance the training convergence by increasing the number of positive query samples. Unlike simply employing one-to-one and one-to-many matching label assignments, our method features a sequential matching process that constructs a **new decoder** LDD. This design is based on experiments and an analysis of the roles of SA and CA (as outlined in Sec. 3.1.3) and addresses the need to transition from a fixed-query to a free-query detector.

In this paper, our primary goal is not to *accelerate model convergence as these methods do*. Instead, we address the **matching ambiguity** that arises from the mixed use of one-to-one and one-to-many matching—**a problem that these methods have overlooked and not effectively resolved**. We tackle this issue at the label assignment level by designing the LDD decoding structure with one-to-many label assignment in BLP and one-to-one label assignment in DP, which *decouples* the two types of matching while enabling each to *fulfill* its role effectively.

At the same time, we explore *another dimension* of one-to-one and one-to-many method (*beyond the label assignment level*), specifically the opposing effects of CA and SA (with CA used to aggregate predictions of boxes to a single object (one object to many queries); SA used to disperse boxes for a single query to one object (one query to one object) and reduce the confidence scores of similar

queries, as detailed in Sec. 3.1.3. By effectively leveraging the CA and SA in decoder structure, we have alleviated the "**recurrent shifting**" issue associated with both CA and SA. This is illustrated in Appendix Table 12 (the ablation on BLP and DP layers) and Table 14 (the ablation on the order of CA and SA), which highlight our "sequential matching" impact on accuracy.

**Differences between H-DETR, MS-DETR and QFree-Det on Sequential Matching.**   Specifically, our work differs from H-DETR Jia et al. (2023) and MS-DETR Zhao et al. (2024) in four main aspects: **(1) Problem Addressed:** The primary goal of both H-DETR and MS-DETR is to enhance the model's *training efficiency* to speed up the training convergence. In contrast, our sequential matching model focuses on resolving "**detection ambiguity**" caused by mixed label assignments (i.e., combining one-to-one and one-to-many) and addressing the "**recurrent shifting**" issue that arises from the interaction between CA and SA. (2) **Design of Decoder Structure:** H-DETR uses *additional* branches to learn one-to-many assignments, which significantly increases the training cost. The MS-DETR shares a *similar* decoder structure as DINO, and introduces *additional heads* (box and class predictors) for one-to-many supervised to further enhance the training efficiency. Different from these two methods, our approach employs a **single** branch and implements an efficient end-to-end structure by dividing the decoder into **decoupled** locating and recognition stages: BLP for localization and DP for refining and de-duplicating detection boxes. Our approach effectively alleviates the "detection ambiguity" from mixing label assignments to *sequential label assignments*, incorporating with the optimized decoder structure by leveraging the unique characteristics of CA and SA to stop the "*recurrent shifting*" problem, *not only ensuring faster convergence, reducing the complexity, but also further mitigating the "detection ambiguity" at the same time.* (3) **Detection Capability:** H-DETR and MS-DETR are both limited to detecting a **fixed** number of objects, whereas our decoder is designed to detect an **adaptive** number of objects, which is beneficial for many applications, such as sparse/dense/open-ended detection tasks. (4) **Detection Accuracy:** Compared to H-DETR, as shown in Table 3, with the same backbone (Swin-T), our model achieves higher performance, increasing by **+2.1%** AP (52.7% vs. 50.6%) and **+2.9%** $AP_S$ (36.3% vs. 33.4%). Compared to MS-DETR, as shown in the Table 3, with the same backbone (ResNet50), our model also achieves higher performance, increasing by **+0.2%** AP (50.5% vs. 50.3%) and **+1.6%** $AP_S$ (34.3% vs. 32.7%) than MS-DETR, further confirming our model's effectiveness. As mentioned in MS-DETR, we also believe that the one-to-many supervision using additional head modules in MS-DETR is a *complementary approach* to our model, which could *potentially* further enhance the training efficiency and accuracy of our method.

### E.3   THE IMPACT OF REMOVING PQ AND THE LDD FRAMEWORK FOR ADDRESSING THE DETECTION AMBIGUITY ISSUE

**The impact of removing PQ.**   Our experiments (Table 1 and Table 13) show that removing PQ has a slight effect on the model's accuracy. We believe this is primarily due to the *interaction mechanisms* of CA and SA in the decoder, as well as *the specific role of PQ* for the decoder.

In original DETR Carion et al. (2020), PQ was *randomly initialized and learned to increase the differences between query embeddings* (as outlined in Sec.3.2 of DETR Carion et al. (2020) paper). The follow-up works adopted a similar query structure to DETR, referring to them as content queries (CQ) and positional queries (PQ) (as mentioned in the Sec. 1 of DAB-DETR Liu et al. (2022)). We have discussed the role of CQ and PQ in the Sec. 3.1.2 and Sec. A.1. With the development of PQ reformulating the box coordinates into PQ embeddings, we can observe that PQ provides the essential object location information for CQ (via plus operation).

However, instead of the static random initialization in DETR, obtaining adaptive query directly from encoder tokens would *inherently contain these object location information*. Specifically, these queries integrate the token information of the regions where the object itself is located, which implicitly contain bounding box positions or offset information at each layer of the decoder, *supervised by the ground truth boxes*. This is one of the reasons why adding additional PQ positional information has a minimal impact on model accuracy.

Additionally, the interaction mechanisms of CA and SA eliminate the need to explicitly include PQ in queries. CA employs deformable attention Zhu et al. (2020) to interact information between the query and encoder features. By inputting the bounding box position of the current query, *CA samples points around this bounding box to interact with the query*, effectively updating the query's information, so that reducing the need for additional positional priors to indicate the object's location. For SA, the queries are derived from encoder tokens, which *contain inherent differences in information between different objects*. This enables the SA module to learn these differences without needing additional positional priors.

**LDD framework for addressing Matching Ambiguity.** In the Sec. 1, Sec. 2, and Sec. 3.1.3, we have discussed that "detection ambiguity" arises from two main reasons: one-to-one and one-to-many label assignments with shared decoder weights, and the opposing roles of CA and SA with "*recurrent shifting*" operation. To address this, we designed a unified LDD framework that effectively decouples mixing label assignments and explicitly removes the "recurrent shifting" operation.

Specifically, the BLP module contains only CA module and employs only the one-to-many supervision mechanism, while the DP module incorporates multiple SA and employs the one-to-one matching label assignment mechanism. These designs effectively eliminate the sources of detection ambiguity from the outset.

When connecting the BLP and DP modules, the conflict between the two label assignments still exists, due to the query is sequentially updated by the BLP and DP modules. To address this, we designed a simple yet effective method for Stopping Gradient back-propagation of Query (SGQ) from DP to BLP, which helps mitigate conflicts between the two modules. As shown in Table 16, the absence of the SGQ approach resulted in a **2.0%** AP drop in the model's performance, highlighting the significance of our approach in alleviating detection ambiguity and further validating LDD's effectiveness.

### E.4 The Computational Complexity Analysis

In this section, we discuss the computational complexity analysis in four aspects:

(1) For the *computational complexity*, we have included analysis in Sec. A.5, indicating that our model can effectively reduce the number of queries, and the corresponding FLOPs are also reduced, as illustrated in Fig. 4.

(2) For the *efficiency* comparison, we have conducted experiments on both COCO and WiderPerson datasets to show the relations between the number of objects, adaptive query numbers, and the corresponding performance (Sec. B.2). The results (in Table 9, Fig. 5) indicate that our model obtained overall higher performance across all subsets of COCO than DINO. Especially for the subset of 1-5, our model only uses **7.25%** of the queries (65.22 vs 900) while achieving higher performance by **+0.9%** AP. For the more challenging dataset of WiderPerson, our model obtains overall higher performance under both sparse and dense scenes (in Table 10, Fig. 7). These results clearly demonstrate the effectiveness of our method.

(3) For the inference time, we conducted *additional* comprehensive tests on inference speed (in Sec. D), varying the number of queries from 10 to 1800. The results indicate that, with the same number of queries as DINO (900 queries), our LDD decoder framework has improved the inference speed of the model's decoder by **+34.8%** over DINO.

(4) In terms of *inference* memory usage, our model is similar to DINO. During inference, modules like CA and SA generate *intermediate variables*, causing dynamic changes in GPU memory. We test and record the **maximum memory allocation** (batch size = 1): 912.3 MB for DINO (900 queries) and 914.9 MB for QFree-Det (900 queries), indicating only a small difference between the two models. This slight variation may be due to differences in the implementation of decoder structures, such as the intermediate variables within the code. Since the adaptive number of queries learned by the model is limited to a small range, the differences in memory usage are minimal. It is noteworthy that the significance of the model's adaptive free number of queries, derived from the

image itself, lies in *its ability to detect a flexible number of objects while reducing computational load*, as illustrated in Sec. A.5.

**Computational resources for dynamic query selection of AFQS algorithm.** For the dynamic query selection, the AFQS algorithm introduces a simple yet effective threshold-based method, which maintains consistent complexity. Specifically, the AFQS algorithm serves two main purposes: converting fixed queries into free queries and addressing the high training GPU memory demand issue associated with excessive query numbers during *training*. It consists of two steps:

(1) generating classification scores for all encoder tokens, which has a *fixed* computation complexity;

(2) selecting queries using a threshold-based method that compares the scores of all encoder tokens to get a *global* solution, also with *fixed* computation complexity.

Additionally, regarding inference time, most of the duration is spent on step one (approximately 7ms for a $2080 \times 800$ image), while step two takes less than **1 ms**.

**How does the model scale with increasing image complexity and object density?** In the Sec. B.2, we have tested the proportional relation between the number of objects and the number of queries. The results show that as the number of objects in an image increases, the rate at which the number of queries increases slows down. To better illustrate this, we calculated the ratio of the model's queries to the number of objects in the image, presented in Table 9. From the table, it can be observed that as the number of objects increases, the growth rate of queries *slows down* and gradually declines, falling from 25.08 to 16.37. Concurrently, our model demonstrates a *growing advantage* over DINO as the number of objects increases across the subsets (31–35, 36–40, 41–45, 46+), as illustrated in Fig. 5. Notably, the subset of 1-5 objects accounts for 55.9% of the images in the val2017 set of COCO. For this subset, our model utilizes only **7.25%** of the queries (65.22 vs. 900) while achieving a higher performance with a **+0.9%** AP, suggesting that the queries selected via AFQS are more effective.

In terms of complexity, we have discussed the changes in FLOPs with varying queries in Sec. A.5. As shown in Fig. 5, the reduction in the number of queries (cyan dashed line) effectively lowers the model's FLOPs (purple solid line). These experiments underscore the effectiveness of our method's adaptive characteristics in object detection.

**Trade-offs between performance and computational cost in QFree-Det.** For the model with an adaptive free number of queries, computational complexity and performance are related to the number of objects in the test images. In the Sec. A.5, we have conducted ablation studies on the classification score $S$ using the COCO dataset, exploring how the number of queries and FLOPs change with variations in the classification score $S$. From Table 11 and Fig. 4, we can observe that a lower threshold $S$ leads to a higher number of queries, which improves the model's accuracy but also increases its computational complexity. When $S$ ranges from 0.01 to 0.05, the model's performance varies in a small range of 50.2% ~ 50.6% AP, indicating good robustness. To balance the trade-offs between performance and computational cost, we set $S = 0.02$ for the configuration used in other experiments in the main paper. With this configuration, our model achieves higher performance (50.5% vs. 49.0% in AP) while maintaining smaller computational complexity (273G vs. 279G) and faster decoder inference speed (9.2ms vs. 14.1ms) than DINO.

## E.5 THE CONTRIBUTIONS OF AFQS ALGORITHM

The decoder query plays a crucial role in reasoning about the relations of the object and the global image context to output the detections, connecting the components of the decoder and greatly impacting the performance. Transforming a fixed-query detector into a free-query detector is *not merely a matter of switching from a predefined to a dynamic number of queries in DETR models*; it is a complex process. As discussed in Sec. 3.1.2 and Sec. 3.1.3, this transformation is influenced

by the *decoder architecture, the statically initialized queries, and the using of self-attention module*. Specifically, the proposed AFQS algorithm is novel in three key aspects:

(1) **New Query Type with Object Information**: AFQS introduces a new query type that alters the existing query composition in DETR models (from content and positional queries to SADQ). This effectively leverages encoder token information and reduces the need of random embeddings for query initialization. Most existing methods direct follow the query design of DETR, where the positional query is original adopted to *increase the difference between quries to produce different results* (as outlined in Sec.3.2 of DETR Carion et al. (2020)). However, our experiment and analysis indicate that the positional query is unnecessary, which is an important insight for future researches on both the fixed-query and free-query detectors.

(2) **Adaptive Queries**: AFQS switches from a fixed, predefined number of queries to an adaptive number of queries, using a simple yet effective threshold-based method to filter appropriate encoder tokens. As the model training progresses, this method gradually enables the classification head to distinguish between positive and negative samples among all encoder tokens, obtaining a global solution.

(3) **Addressing Training GPU Memory Limitation for Using SA module**: AFQS effectively addresses training GPU memory limitations, allowing for training the model with the essential self-attention (SA) module within the decoder architecture. In contrast, directly switching from fixed queries to free queries would significantly degrade performance (e.g., a **4.9%** AP drop as shown in Table 2).

### E.6 Advantages of the QFree-Det Model

The primary limitation of *query-fixed* detectors is their requirement to predict a **large** and **fixed** number of detection results for both **sparse** and **dense** scenes. On one hand, this results in a considerable amount of *unnecessary and redundant computation*. On the other hand, when these fixed-query detectors are applied to more challenging downstream tasks, such as Open-Ended Detection Lin et al. (2024), these redundant queries would be further fed into the **large language model** (LLM) to generate object names directly, which will significantly increase the overall computational complexity and cost, as illustrated in Sec. D.

In contrast, our *query-free* detector *adaptively* eliminate redundant queries at *an early stage* in the transformer-based detector. This leads to a **more efficient, cost-effective, accurate, and flexible approach**, as demonstrated in the following four aspects:

(1) **High-rate of effective query utilization, better cost-efficiency, and higher performance.** In *sparse scenarios*, QFree-Det achieves comparable or even higher accuracy with only a small number of queries, while simultaneously reducing computational load of the decoder. The transformer-decoder is primarily composed of layers of cross-attention (CA) and self-attention (SA). CA has a complexity of $o(NKC^2)$ (using deformable attention Zhu et al. (2020), where $N$ is the number of queries, K is the number of sample points, and $C$ denotes the number of channels), while SA has a complexity of $o(N^2)$. As the number of queries $N$ decreases, the computational load of the CA and SA reduces at a linear and quadratic rate, respectively. This makes the approach *highly cost-effective* and results in *lower power consumption* during deployment, particularly in *edge devices*.

In many scenarios, objects within the image are often sparse. For instance, in the val2017 of COCO dataset, **55%** of images contain between 1 and 5 objects (as shown in Sec. B.2, Table 9). In this context, we achieved an accuracy that is **+0.9%** AP higher than the DINO model while using only **7.25%** of the queries (65.22 vs. 900), clearly demonstrating the effectiveness of our query with high-rate utilization.

(2) In dense scenarios, our query-free model also achieves higher accuracy with fewer queries than DINO. This robust advantage becomes even more pronounced as the number of objects increases, as illustrated in Fig. 5 and Table 9 (specifically in the subsets of $31 \sim 35, 36 \sim 40, 41 \sim 45, 46+$).

(3) For more complex *multi-modal* detection tasks, such as open-ended detection Lin et al. (2024), decoder queries are inputted into **large language models** to directly generate corresponding object names without additional vocabulary priors. This process is highly **complexity-sensitive** to the

number of *queries* (for instance, the LLM model shows 10,139.64 GFLOPs with 900 queries and 2,253.45 GFLOPs with 200 queries Lin et al. (2024)). However, fixed-query approaches that using a large fixed number of queries significantly increase the computational load. In contrast, our query-free method reduces this redundancy by eliminating unnecessary queries at an early stage, which is highly significant and holds great potential for these multi-modal tasks.

(4) Our query-free method offers greater **flexibility** through its inherent adaptive characteristics. For example, once the model is trained, the number of queries can be adaptively adjusted to suit *various scenarios* without the need for retraining, which is still required for fixed-query detectors.

Furthermore, our model has demonstrated superior performance on the COCO dataset (Table 3), as well as on the more challenging WiderPerson (Table 4) and CrowdHuman (Table 19) datasets than the DINO model. This further indicates its *robustness* across a variety of scenarios.

In addition, we also introduce a novel decoding framework, LDD, which effectively tackles the issue of "detecting ambiguity" caused by mixing label assignments of one-to-one and one-to-many, as well as the "recurrent shifting" problem. The effectiveness of our decoding framework is demonstrated by the experimental results in Table 12 and Table 16. It *enhances* performance while **simplifying** the decoder architecture by using *fewer* SA layers, a single adaptive query type, and *eliminating the need for additional branches or multi-head prediction modules*.

Finally, our LDD framework significantly improves the speed of the decoder by **+34.8%** compared to DINO (as detailed in Sec. D). Although the overall speed advantage may not be substantial due to the multiple components of transformer-based detectors——where the slow inference speed is a common bottleneck, especially when compared to faster CNN-based detectors like YOLO Khanam & Hussain (2024))—— it offers a promising solution for enhancing the speed of transformer-based detectors. For instance, by integrating the YOLO *backbone*, the Sparse-DETR Roh et al. (2021) *encoder*, and our LDD *decoder*, we may have great potential to develop a *high-speed, low-cost, high-performance, query-free* transformer detector, which deserves further exploration in future work.

