# OpenReview forum: "QFree-Det: Query-Free Detector with Transformer and Sequential Matching"
_ICLR.cc/2025/Conference — Submitted to ICLR 2025_

### Official Review · Reviewer_eExB · 2024-11-01

**Soundness:** 3
**Presentation:** 3
**Contribution:** 1
**Rating:** 5
**Confidence:** 5

**Summary:**

Transformer-based detectors like DETR and DINO have a fixed capacity for detecting objects, leading to missed detections and false positives. To overcome these limitations, QFree-Det is proposed as a dynamic, query-free detector that adapts to varying object counts in input images. It features an Adaptive Free Query Selection (AFQS) algorithm for flexible query selection and a Location-Deduplication Decoder (LDD) that separates one-to-one and one-to-many detection processes to reduce ambiguities. Additionally, a unified PoCoo loss enhances small object detection.

**Strengths:**

1. The idea of producing a flexible number of queries to predict a flexible number of detections, is interesting.
2. The paper is easy to follow, and the presentation is good.
3. The performance of the proposed QFree-Det compared to SOTAs looks good.

**Weaknesses:**

1. The contribution of AFQS is not strong enough. The main idea of the proposed Adaptive Free Query Selection (AFQS) algorithm: The fixed content query (CQ) is replaced by the proposed flexible self-adaptive decoder query (SADQ), where SADQ is obtained by sorting the classification scores of all N encoder tokens and selecting the scores above a certain threshold as M SADQ. In short, the AFQS algorithm is a simple strategy to reduce N queries into M queries by sorting the classification scores. I don't think it is enough to be a main contribution.
2. The novelty of sequential matching is incremental, where sequential matching combines Box Locating Part (One-to-many matching), and Deduplication Part (One-to-one matching). One-to-many matching and One-to-one matching are the existing methods.
3. The framework of Box Locating Part (One-to-many matching) and Deduplication Part (One-to-one matching), looks complex. Deduplication Part (One-to-one matching) can be replaced by the parameter-free NMS. The Deduplication Part (One-to-one matching) looks more complex than NMS.
4. PoCoo loss is useful for small object detection. However, the connection between PoCoo loss and the motivation of designing a query-free detector, is very weak.
5. Dynamic query selection and the dual-step decoding process may require more computational resources, potentially impacting processing speed in real-time detection applications.
6. According to Table 5, free-query with AFQS only is worse than fixed-query.

**Questions:**

see the weaknesses.

---

> ### Author Response · Authors · 2024-11-20
> **Responses to the Reviewer eExB (part 1)**
>
> Dear reviewer,
>
> We sincerely appreciate your thoughtful and constructive feedback. We are eager to respond the concerns and hope our responses will align with your expectations.
>
>
> **Q1: The contribution of AFQS algorithm.**
>
> We appreciate for the valuable feedback and suggestions. The decoder query plays a crucial role in reasoning about the relations of the object and the global image context to output the detections, connecting the components of the decoder and greatly impacting the performance. Transforming a fixed-query detector into a free-query detector is *not merely a matter of switching from a predefined to a dynamic number of queries in DETR models*; it is a complex process. As discussed in **Sec.3.1.2** and **Sec.3.1.3** of the main paper, this transformation is influenced by the *decoder architecture, the statically initialized queries, and the using of self-attention module*.
>
> Specifically, the proposed AFQS algorithm is novel in three key aspects:
>
> (1) **New Query Type with Object Information**: AFQS introduces a new query type that alters the existing query composition in DETR models (from content and positional queries to SADQ). This approach effectively leverages encoder token information and reduces the need of random embeddings for query initialization. Most existing methods direct follow the query design of DETR, where the positional query is original adopted to *increase the difference between quries to produce different results* (as outlined in **Sec.3.2** of DETR [1] paper). However, our experiment and analysis indicate that the positional query is unnecessary (please see details in our response to the Reviewer **TZb2** of **Q1.1**), which is an important insight for future researches on both the fixed-query and free-query detectors.
>
> (2) **Adaptive Queries**: AFQS switches from a fixed, predefined number of queries to an adaptive number of queries, using a simple yet effective threshold-based method to filter appropriate encoder tokens. As the model training progresses, this method gradually enables the classification head to distinguish between positive and negative samples among all encoder tokens, obtaining a global solution.
>
> (3) **Addressing Training GPU Memory Limitation for Using SA module**: AFQS effectively addresses training GPU memory limitations, allowing for training the model with the essential self-attention (SA) module within the decoder architecture. In contrast, directly switching from fixed queries to free queries would significantly degrade performance (e.g., a **4.9\%** AP drop as shown in **Table 2** of the main paper).
>
> [1] Carion N, Massa F, Synnaeve G, et al. End-to-end object detection with transformers[C]// ECCV2020.
>
> ---
>
> **Q2: Sequential matching is incremental: One-to-many matching and One-to-one matching are the existing methods**.
>
> One-to-one matching and one-to-many matching label assignments have been demonstrated in several studies (such as H-DETR, Align-DETR, and DAC-DETR, etc) to enhance the training convergence by increasing the number of positive query samples. In Appendix **Sec.A.4** of the main paper, we have discussed the differences between our method and existing one-to-many methods in terms of motivation, implementation, and performance.
> Unlike simply employing one-to-one and one-to-many matching label assignments, our method features a sequential matching process that constructs a **new decoder**. This design is based on experiments and an analysis of the roles of SA and CA (as outlined in **Sec.3.1.3** of main paper) and addresses the need to transition from a fixed-query to a free-query detector. For more discussion about ''*Sequential Matching*'', please see our response to Reviewer **rMNn** of **Q1**.
>
> Furthermore, our goal is NOT to *accelerate model convergence as these methods do*. Instead, we address the ``matching ambiguity`` that arises from the mixed use of one-to-one and one-to-many matching—*a problem that these methods have overlooked and not effectively resolved*. We tackle this issue at the label assignment level by designing the LDD decoding structure with one-to-many label assignment in BLP and one-to-one label assignment in DP, which *decouples* the two types of matching while enabling each to *fulfill* its role effectively.

---

> ### Author Response · Authors · 2024-11-20
> **Responses to the Reviewer eExB (part 2)**
>
> **Continue the answer of part 1 to Q2:**
>
> At the same time, we explore *another dimension* of one-to-one and one-to-many method (*beyond the label assignment level*), specifically the opposing effects of CA and SA (with CA used to aggregate predictions of boxes to a single object (one object to many queries); SA used to disperse boxes for a single query to one object (one query to one object) and reduce the confidence scores of similar queries, as detailed in **Sec.3.1.3** of the main paper). By effectively leveraging the CA and SA in decoder structure, we have alleviated the ``recurrent shifting`` issue associated with both CA and SA. This is illustrated in Appendix **Table 12** (the ablation on BLP and DP layers) and **Table 14** (the ablation on the order of CA and SA) of the Appendix in main paper, which highlight our ''*sequential matching*'' impact on accuracy.
>
> ---
>
> **Q3: Deduplication Part (One-to-one matching) can be replaced by the parameter-free NMS. The Deduplication Part (One-to-one matching) looks more complex than NMS**.
>
> Thanks very much for the significant comment. DP is an integral part of the decoder and not a post-processing algorithm. Although we designed DP to eliminate duplicates, it is distinct from NMS and cannot be directly replaced by it. Our model's decoder, similar to those in existing DETR-like models, comprises six layers: 4 LBP layers and 2 DP layers, connected in series. As analyzed in original DETR [1] paper, the decoder primarily reasons about the relations of queries and image context to generate detections, which is *necessary* for the detector. In **Table 12** of the Appendix of main paper, we conducted ablations on the BLP and DP layers. The results (*line 941* of main paper) indicate that reducing the number of DP layers (for example, using only one DP layer) is harmful for the performance, underscoring the importance of the DP layer.
>
> In **Table 2** (line 249 and 250) of the main paper, we present a *free-query* model that removes all SA (approximating the removal of the DP module), while employing the NMS to eliminate duplications. This model achieved only 47.5% AP on COCO, which is significantly lower than the 50.5% AP achieved with our model with DP, clearly highlighting the importance and irreplaceability of the DP.
>
> On the other hand, NMS can only perform deduplication on the detection results that have already been obtained, and it is sensitive to parameters; it cannot generate detection results directly, making it unsuitable as a replacement for the decoder. The DETR model was originally designed to simplify the detection process in an **end-to-end** pipeline, removing post-processing steps like NMS and improving model robustness. *Our method aligns well with this goal*, enabling the model to function effectively without relying on NMS, but not to pull NMS back again.
>
> The original DETR paper also notes that the FFN (Feed Forward Network, FFN) in the decoder has a significant impact on model accuracy. We have made a test on FFN of DINO model to reduce computational complexity by decreasing the hidden layer dimension in the FFN from 2048 to 768. However, this change resulted in a **1.6\%** decrease in AP, underscoring the critical role of the decoder network layers in maintaining performance. This further underscores the importance of the decoder.
>
> [1] Carion N, Massa F, Synnaeve G, et al. End-to-end object detection with transformers[C]// ECCV2020.
>
> ---
>
> **Q4: PoCoo loss is useful for small object detection. However, its connection to the motivation of designing a query-free detector is very weak**.
>
> Thanks, we agree with it, and PoCoo loss is a supplementary contribution of our method. As presented in Sec.3.5 of main paper, PoCoo loss is designed based on IA-BCE loss, and IA-BCE loss efficiently addressed the correlation between classification score and localization precision. Our PoCoo loss further effectively leverages the prior information of box annotation, playing a crucial role in the model's training process.

---

> ### Author Response · Authors · 2024-11-20
> **Responses to the Reviewer eExB (part 3)**
>
> **Q5: Dynamic query selection and the dual-step decoding process may require more computational resources, potentially impacting processing speed in real-time detection applications**.
>
> In QFree-Det, the dynamic query selection and the dual-step decoding process do not require substantial additional computational resources.
>
> **For the dynamic query selection**, the AFQS algorithm introduces a simple yet effective threshold-based method, which maintains consistent complexity. Specifically, the AFQS algorithm serves two main purposes: converting fixed queries into free queries and addressing the high training GPU memory demand issue associated with excessive query numbers during *training*. It consists of two steps:
>
> (1) generating classification scores for all encoder tokens, which has a *fixed* computation complexity;
>
> (2) selecting queries using a threshold-based method that compares the scores of all encoder tokens to get a *global* solution, also with *fixed* computation complexity.
>
> Additionally, regarding inference time, most of the duration is spent on step one (approximately 7ms for a 2080 $\times$ 800 image), while step two takes less than **1 ms** (For more details, please refer to our responses in the **General Responses** section).
>
>
> **For the dual-step decoding process**, as mentioned in the responses to **Q3** and **Q6**, our LDD decoder consists of 4 LBP layers and 2 DP layers (totaling 6 CA and 4 SA layers), reducing both the number of parameters and computational load compared to existing methods, which use 6 decoding layers (with 6 CA and 6 SA layers). We further tested the model's inference speed, and our LDD decoder is faster than the decoder of DINO, with an improvement of **34.8\%** under the same number of queries (more details in **General Response**), indicating the effectiveness of our AFQS method and LDD framework.
>
> Transformer-based detectors commonly face challenges in *real-time* applications due to their core multiple components, which involve numerous time-consuming attention operations. Our newly designed efficient LDD decoder framework helps mitigate this issue to some extent, making it possible to integrate with lightweight models like YOLO and further enhance the model's speed. Additionally, our free-query model holds significant *potential* for open-ended object detection task. This task directly generates object names through a generative large language model, where the query count greatly influences the model's complexity. The adaption characteristic of our model enables reduced queries to feed into generative large language model, thereby decreasing the workload for subsequent category generation processing, which highlights its potential applications and importance for the future research in vision community.
>
> ---
>
> **Q6: According to Table 5, free-query with AFQS only is worse than fixed-query**.
>
> Yes, the performance of the free-query approach using only AFQS is slightly below (by *0.3%* AP) than fixed-query method. Naturally, within an appropriate range of query numbers, *more queries can lead to more detections*, thereby enhancing model accuracy. A fixed number of predefined queries (such as 900) maintains a constant query count across both sparse and dense scenarios. In contrast, when applying the AFQS algorithm alone, the number of queries adapts to the input image, which presents a *greater challenge* for SA, as it must accommodate various patterns of query numbers. This may lead to a *slight decrease* in accuracy, as shown in Table 5 (line 477 in the main paper). However, compared to the fixed-query detector, the free-query detector is crucial for enhancing model's flexibility, applicability, and scalability in real-world scenarios, making it a promising trend for future research. Furthermore, the issue of accuracy decrease can be effectively mitigated through other methods.
>
> The decrease in accuracy resulting from the reduction in the number of queries is a fundamental limitation of the query-based detection framework,
> where each object prediction is from a single query. This issue is hard to be
> effectively addressed by relying *solely* on query selection algorithms like AFQS. As a result, we introduce a more efficient decoder of LDD architecture, which incorporates 2 SA units in each DP module, enhancing the ability of decoder to handle different patterns of query numbers. After using the LDD decoder, the accuracy of our model improved by **+1.2%** AP, as shown in Table 5 (line 478) of the main paper.
>
> Our LDD architecture effectively leverages the roles of CA and SA, enabling the DP module to adapt to these pattern changes of query numbers. In addition, the overall number of SA used in our decoder is two fewer than in existing methods (4 vs. 6), and there are *NO additional decoder branches* in LDD architecture compared to other methods, further demonstrating the effectiveness of our AFQS and LDD approaches.

---

> > ### Author Response · Authors · 2024-12-03
> > **Kindly remainder for the feedback in the final discussion phase**
> >
> > Dear Reviewer eExB,
> >
> > We sincerely appreciate your valuable time, diligent efforts, and thorough reviews. We hope that the detailed clarifications (Part 1, Part 2, and Part 3) in our responses, along with the revised manuscript, address your concerns. If all issues have been resolved, we would be grateful if you could kindly reconsider the score based on the revised manuscript and all clarifications. We highly value your thoughtful and constructive feedback, and we are dedicated to improving our paper to align with your expectations.
> >
> > Thank you once again for your time and consideration.
> >
> > Best regards,
> >
> > Authors

---

### Official Review · Reviewer_rMNn · 2024-11-03

**Soundness:** 3
**Presentation:** 3
**Contribution:** 3
**Rating:** 5
**Confidence:** 3

**Summary:**

This paper propose QFree-Det, a query-free detector that select dynamic queries from encoders and decouples one-to-one and one-to-many matching in sequential way. It achieves remarkable improvements compared to the DINO baseline.

**Strengths:**

1. The paper analyze the ambiguity problem overlooked in current one-to-many detectors, and propose a new sequential matching method to tackle this problem, which is not explored before.
2. The paper is well-written and the experiments are comprehensive.

**Weaknesses:**

The dynamic query selection mechanism is similar to the distinct query selection process in DQS[1]. And the sequential matching process is similar to the hybrid layers in H-DETR[2] and MS-DETR[3]. It is better to add citation and discussion with them.

**Questions:**

See the weakness part

---

> ### Author Response · Authors · 2024-11-20
> **Responses to the Reviewer rMNn**
>
> Dear reviewer,
>
> We greatly appreciate your valuable feedback and suggestions, and we have carefully analyzed and compared the differences between these methods, in hopes of effectively addressing the issues.
>
>
>
> **Q1: The sequential matching process is similar to the hybrid layers in H-DETR and MS-DETR**
>
> In the **Sec.A.4** (line 800 to 816) of Appendix in the main paper, we have discussed the difference between our approach with existing one-to-many methods, such as H-DETR [1] and DAC-DETR [2]. And, we previously overlooked the discussion of our work with MS-DETR [3] approach. For convenience, here, we discuss the differences between our method and both two approaches. Following the suggestion, we will include these citation and discussion in the next version of the paper.
>
> Specifically, the sequential matching process in our work differs from H-DETR and MS-DETR in *four* main aspects:
> **(1) Problem Addressed:** The primary goal of both H-DETR and MS-DETR is to enhance the model's ``training efficiency`` to speed up the training convergence. In contrast, our sequential matching model focuses on resolving ``detection ambiguity`` caused by mixed label assignments (i.e., combining one-to-one and one-to-many) and addressing the ``recurrent shifting`` issue (*line 273* in the main paper) that arises from the interaction between CA and SA.
> (2) **Design of Decoder Structure:** H-DETR uses *additional* branches to learn one-to-many assignments, which significantly increases the training cost. The MS-DETR shares a *similar* decoder structure as DINO, and introduces *additional heads* (box and class predictors) for one-to-many supervised to further enhance the training efficiency. Different from these two methods, our approach employs a *single* branch and implements an efficient end-to-end structure by dividing the decoder into **decoupled** locating and recognition stages: BLP for localization and DP for refining and de-duplicating detection boxes. Our approach effectively alleviates the ''detection ambiguity'' from mixing label assignments to *sequential label assignments*, incorporating with the optimized decoder structure by leveraging the unique characteristics of CA and SA to stop the ''*recurrent shifting*'' problem, *not only ensuring faster convergence, reducing the complexity, but also further mitigating the ''detection ambiguity'' at the same time*.
> (3) **Detection Capability:** H-DETR and MS-DETR are both limited to detecting a *fixed* number of objects, whereas our decoder is designed to detect an *adaptive* number of objects, which is beneficial for many applications, such as sparse/dense/open-ended detection tasks.
> (4) **Detection Accuracy:** Compared to H-DETR, as shown in **Table 3** (line 399) of the main paper, with the same backbone (Swin-T), our model achieves higher performance, increasing by **+2.1\%** AP (52.7\% vs. 50.6\%) and **+2.9\%** AP$_S$ (36.3\% vs. 33.4\%). For your convenience, we list it as below:
>
> | Model | Backbone | Objects | Epochs | AP | AP$_{50}$ | AP$_{75}$ | AP$_S$ | AP$_{M}$ | AP$_{L}$ |
> | :---: | :---: | :---: | :---: | :---: | :---: | :---: | :---: | :---: | :---: |
> | H-DETR | Swin-T | fixed | 12 | 50.6 | 68.9 | 55.1 | 33.4 | 53.7 | 65.9 |
> | **QFree-Det (ours)** | Swin-T | *free* | 12 | **52.7 (+2.1)** | **70.2** | **57.6** | **36.3 (+2.9)** | **56.4** | **67.7** |
> | MS-DETR | ResNet50 | fixed | 12 | 50.3 | 67.4 | 55.1 | 32.7 | 54.0 | **64.6** |
> |**QFree-Det (ours)** | ResNet50 | *free* | 12 | **50.5 (+0.2)** | **67.5** | 55.1 | **34.3 (+1.6)** | **54.6** | 64.5 |
>
>
>
> Compared to MS-DETR, as shown in the above table, with the same backbone (ResNet50), our model also achieves higher performance, increasing by **+0.2\%** AP (50.5\% vs. 50.3\%) and **+1.6\%** AP$_S$ (34.3\% vs. 32.7\%) than MS-DETR, further confirming our model's effectiveness. As mentioned in MS-DETR, we also believe that the one-to-many supervision using additional head modules in MS-DETR is a *complementary approach* to our model, which could *potentially* further enhance the training efficiency and accuracy of our method.
>
>
> [1] Jia D, Yuan Y, He H, et al. Detrs with hybrid matching[C]//CVPR 2023.
>
> [2] Hu, Z, Sun, Y, Wang, J, et al. DAC-DETR: Divide the attention layers and conquer[C]//NIPS 2023.
>
> [3] Zhao C, Sun Y, Wang W, et al. MS-DETR: Efficient DETR Training with Mixed Supervision[C]//CVPR 2024.
>
> ---
>
> **Q2: The dynamic query selection mechanism is similar to the distinct query selection process in DQS.**
>
> Thank you very much for your comment. We apologize, but due to the lack of reference, we cannot clearly identify which specific paper needs to be compared. We are unsure if the mentioned DQS method refers to the DDQ [1] approach. If that is the case, please refer to our response to Reviewer **Pjik** regarding **Q1**. Thank you very much.
>
> [1] Zhang S, Wang X, Wang J, et al. Dense distinct query for end-to-end object detection[C]//CVPR 2023.

---

> > ### Author Response · Authors · 2024-12-03
> > **Kindly remainder for the feedback in the final discussion phase**
> >
> > Dear Reviewer rMNn,
> >
> > We greatly appreciate your valuable feedback and suggestions. Following
> > your recommendations, we have carefully analyzed and compared the differences between these methods and further revised the manscript accordingly. We hope your concerns are addressed. If all issues have been resolved, we would be grateful if you could kindly reconsider rating based on the revised manuscript and our clarifications. We truly value your thoughtful and constructive feedback, and we are committed to improving our paper to align with your expectations.
> >
> > Thank you once again for your time and consideraton.
> >
> > Best regards,
> >
> > Authors

---

### Official Review · Reviewer_TZb2 · 2024-11-05

**Soundness:** 3
**Presentation:** 2
**Contribution:** 3
**Rating:** 6
**Confidence:** 4

**Summary:**

This paper proposes QFree-Det, a novel query-free detector based on transformers, aiming to address the limitations of fixed query numbers and detection ambiguity in existing transformer-based detectors. The main contributions include the Adaptive Free Query Selection (AFQS) algorithm and the Location-Deduplication Decoder (LDD) framework.

**Strengths:**

-The AFQS algorithm and LDD framework provide a novel solution to the challenges of fixed query limitations and detection ambiguities that are common in traditional transformer-based detectors like DETR and DINO.

-The model achieves state-of-the-art results on the COCO2017 and WiderPerson datasets, particularly excelling in small object detection. This demonstrates the effectiveness of the proposed method over existing leading techniques.

-By enabling  dynamic query selection, QFree-Det effectively adapts to varying scene complexities, providing a robust framework for diverse detection scenarios.

-The introduction of the PoCoo loss function significantly enhances the detection capabilities for small objects, addressing a common challenge in object detection.

**Weaknesses:**

-The authors propose AFQS algorithms and LDD frameworks, but lack in-depth theoretical analysis.  There is no detailed explanation of why removing location queries (PQ) does not significantly affect model performance, nor is there a theoretical proof of how LDD can effectively solve the detection ambiguity problem.

-Insufficient comparison with state-of-the-art methods, In Table 3, the authors mainly compare with older methods such as DINO, but lack detailed comparisons with more recently published methods (e.g. DAC-DETR, Co-DETR, etc.). This makes it difficult to assess the position of QFree-Det in the current research frontier.


-The author mentions that AFQS can reduce redundant queries, but does not provide a detailed computational complexity analysis or efficiency comparison with other methods. Lack of specific inference time and memory usage data.

-Although the improvement of the PoCoo loss function for small object detection is shown in Table 7, there is no in-depth analysis of why this loss function can be particularly effective in improving small object detection performance.

**Questions:**

-How does QFree-Det perform on other challenging datasets beyond COCO and WiderPerson? Are there specific scenarios where QFree-Det might underperform?

-Can you provide a more detailed analysis of the trade-offs between performance and computational cost in QFree-Det?

-How does the AFQS algorithm scale with increasing image complexity and object density?

---

> ### Author Response · Authors · 2024-11-20
> **Responses to the Reviewer TZb2 (part 1)**
>
> Dear reviewer,
>
> We really appreciate your thorough and constructive feedback and suggestions. We have carefully considered your suggestions and made every effort to address the issues well by adding relevant experiments and in-depth analyses.
>
>
> **Q1: In-depth analysis about removing PQ dose not significantly affect model performance and how the LDD framework solve the detection ambiguity problem.**
>
> Thanks for the feedback. These two questions are very worthwhile points to be discussed.
>
> **Q1.1 Removing PQ ...** Our experiments (*Table 1* and *Table 13* of the main paper) show that removing PQ has a slight effect on the model's accuracy. We believe this is primarily due to the *interaction mechanisms* of CA and SA in the decoder, as well as *the specific role of PQ* for the decoder.
>
> In original DETR [1], PQ was *randomly initialized and learned to increase the differences between query embeddings* (as outlined in *Sec.3.2* of the DETR paper). The follow-up works adopted a similar query structure to DETR, referring to them as content queries (CQ) and positional queries (PQ) (as mentioned in the Introduction of DAB-DETR [2]). We have discussed the role of CQ and PQ in the *Sec.3.1.2* of the main paper and *Sec.A.1* of the Appendix. With the development of PQ reformulating the box coordinates into PQ embeddings, we can observe that PQ provides the essential object location information for CQ (via plus operation).
>
> However, instead of the static random initialization in DETR, obtaining adaptive query directly from encoder tokens would *inherently contain these object location information*. Specifically, these queries integrate the token information of the regions where the object itself is located, which implicitly contain bounding box positions or offset information at each layer of the decoder, *supervised by the ground truth boxes*. This is one of the reasons why adding additional PQ positional information has a minimal impact on model accuracy.
>
> Additionally, the interaction mechanisms of CA and SA eliminate the need to explicitly include PQ in queries. CA employs deformable attention [3] to interact information between the query and encoder features. By inputting the bounding box position of the current query, *CA samples points around this bounding box to interact with the query*, effectively updating the query's information, so that reducing the need for additional positional priors to indicate the object's location. For SA, the queries are derived from encoder tokens, which *contain inherent differences in information between different objects*. This enables the SA module to learn these differences without needing additional positional priors.
>
> **Q1.2 LDD framework ...** In the Introduction (*lines 74 to 81*), Related Work (*lines 135 to 145*), and Sec.3.1.3 (*lines 264 to 277*) of the main paper, we have discussed that ``detection ambiguity`` arises from **two** main reasons: one-to-one and one-to-many *label assignments* with shared decoder weights, and the opposing roles of CA and SA with ``recurrent shifting`` (*line 273* in the main paper) operation. To address this, we designed a unified LDD framework that effectively decouples mixing label assignments and explicitly removes the ''recurrent shifting" operation.
>
> Specifically, the BLP module contains only CA module and employs only the one-to-many supervision mechanism, while the DP module incorporates multiple SA and employs the one-to-one matching label assignment mechanism. These designs effectively eliminate the sources of detection ambiguity from the outset.
>
> When connecting the BLP and DP modules, the conflict between the two label assignments still exists, due to the query is sequentially updated by the BLP and DP modules (see details in *Sec.3.4*, lines 367 to 372 of the main paper). To address this, we designed a simple yet effective method for Stopping Gradient back-propagation of Query (SGQ) from DP to BLP, which helps mitigate conflicts between the two modules. As shown in **Table 16** of Appendix in the main paper, the absence of the SGQ approach resulted in a **2.0%** AP drop in the model's performance, highlighting the significance of our approach in alleviating detection ambiguity and further validating LDD's effectiveness.
>
>
> [1] Carion N, Massa F, Synnaeve G, et al. End-to-end object detection with transformers[C]// ECCV2020.
>
> [2] Liu S, Li F, Zhang H, et al. DAB-DETR: Dynamic Anchor Boxes are Better Queries for DETR[C]//ICLR2022.
>
> [3] Zhu X, Su W, Lu L, et al. Deformable DETR: Deformable Transformers for End-to-End Object Detection[C]//ICLR2021.

---

> ### Author Response · Authors · 2024-11-20
> **Responses to the Reviewer TZb2 (part 2)**
>
> **Q2: Comparision with SOTA: DAC-DETR, Co-DETR**
>
> Thanks for the comment. The comparisons of Co-DETR (in *line 389*) and DAC-DETR (in *line 393*) are listed in *Table 3* of the main paper. Under the same backbone and training epochs, our model obtains higher performances than Co-Detr-4scale and DAC-DETR, with improvements of **+1.0\%** (50.5\% vs. 49.5\%) and **+0.5\%** (50.5\% vs. 50.0\%) in AP, **+1.9\%** (34.3\% vs. 32.4\%) and **+1.4\%** (34.3\% vs. 32.9\%) in AP$_S$, respectively. This confirms the effectiveness of our model. For your convenience, we list the results below:
>
>
> | Model | Backbone | Objects | Codebase | Epochs | AP | AP$_{50}$ | AP$_{75}$ | AP$_S$ | AP$_{M}$ | AP$_{L}$ |
> | :---: | :---: | :---: | :---: | :---: | :---: | :---: | :---: | :---: | :---: | :---: |
> | Co-DETR-4scale | ResNet50 | fixed | MMDetection | 12 | 49.5 | **67.6** | 54.3 | 32.4 | 52.7 | 63.7 |
> | DAC-DETR | ResNet50 | fixed | H-DETR-official-public | 12 | 50.0 | **67.6** | 54.7 | 32.9 | 53.1 | 64.2 |
> | **QFree-Det (ours)** |  ResNet50 | *free* | DINO-official-public | 12 | **50.5** | 67.5 | **55.1** | **34.3** | **54.6** | **64.5** |
>
>  ---
>
> **Q3: Computational complexity analysis or efficiency comparison with other methods: inference time and memory usage data.**
>
> Thanks for the comment.
>
> (1) For the computational complexity, we have included analysis in Appendix **A.5** of the main paper, indicating that our model can effectively reduce the number of queries, and the corresponding
> FLOPs are also reduced (in **Fig.4** of Appendix in the main paper).
>
> (2) For the efficiency comparison, we have conducted experiments on both COCO and WiderPerson datasets to show the relations between the number of objects, adaptive query numbers, and the corresponding performance (see Appendix **B.2** in the main paper). The results (in **Table 9, Fig.5**) indicate that our model obtained overall higher performance across all subsets of COCO than DINO. Especially for the subset of 1-5, our model only uses **7.25\%** of the queries (65.22 vs 900) while achieving higher performance by **+0.9\%** AP. For the more challenging dataset of WiderPerson, our model obtains overall higher performance under both sparse and dense scenes (in Appendix **Table 10, Fig.6** of the main paper). These results clearly demonstrate the effectiveness of our method.
>
> (3) For the inference time, we conducted *additional* comprehensive tests on inference speed, varying the number of queries from 10 to 1800. The results indicate that, with the same number of queries as DINO (900 queries), our LDD decoder framework has improved the inference speed of the model's decoder by **+34.8\%** over DINO. For more details, please refer to our responses in the **General Responses** section. On the other hand, our query-free model, which adapts the number of queries based on the image itself, holds significant potential for *open-ended object detection task*. This task directly generates object names through a generative large language model, where the query count greatly influences the model's complexity. The adaption characteristic of our model enables reduced queries to feed into generative large language model, thereby decreasing the workload for subsequent category generation processing, which highlights its potential applications and importance for the future research in vision community.
>
>
>
> (4) In terms of *inference* memory usage, our model is similar to DINO. During inference, modules like CA and SA generate *intermediate variables*, causing dynamic changes in GPU memory. We test and record the *maximum memory allocation* (batch size = 1): 912.3 MB for DINO (900 queries) and 914.9 MB for QFree-Det (900 queries), indicating only a small difference between the two models. This slight variation may be due to differences in the implementation of decoder structures, such as the intermediate variables within the code. Since the adaptive number of queries learned by the model is limited to a small range, the differences in memory usage are minimal. It is noteworthy that the significance of the model's adaptive free number of queries, derived from the image itself, lies in *its ability to detect a flexible number of objects while reducing computational load*, as illustrated in Appendix **A.5** of the main paper.
>
> In addition, the AFQS algorithm serves two main purposes: converting fixed queries into free queries and addressing the issue of high training GPU memory demand associated with excessive query numbers ***during training***, which inhibits the usage of SA (see **Sec.3.1.3** of the main paper). Its primary goal is not to directly reduce memory usage. Furthermore, since the model does not require additional gradient information during inference, the overall memory usage has little impact on the model's inference process, even with a larger number of queries.

---

> ### Author Response · Authors · 2024-11-20
> **Responses to the Reviewer TZb2 (part 3)**
>
> **Q4: In-depth analysis: why PoCoo loss function can be particularly effective in improving small object detection performance.**
>
> Thanks for your valuable comment. PoCoo loss works by integrating additional prior information regarding box sizes in training process. During training , our PoCoo loss *automatically categorizes smaller objects as hard samples based on their annotation information of box size* within the image. Explicitly assigning higher weights to small objects would **up-weight hard samples** and thus focus training on them, which has a similar effect as the data-based method. Specifically, the $[* ]$ term in Eq.(6) of our PoCoo loss, which explicitly introduces the prior of box size information into the loss function, shares a similar motivation with the modulating factor of $(1 - p_t)^\gamma$ in focal loss [1].
>
> [1] Lin T Y, Goyal P, Girshick R, et al. Focal Loss for Dense Object Detection[C]//ICCV2017.
>
> ---
>
> **Q5: Experiments on other datasets? Are there specific scenarios where QFree-Det might underperform?**
>
> **Q5.1 Experiments on other datasets.** For the experiments in the main paper, we have built various variants of QFree-Det on common benchmark datasets of COCO and WiderPerson, with **5** different backbone models across CNN-based, MLP-based, Transformer-based, and Mamba-based backbones, and the large models of Swin-L is also included. The comprehensive experimental results presented in *Table 3,4 and other extensive ablation studies* in the main paper have demonstrated the effectiveness and superiority of our QFree-Det approach.
>
> However, as suggested, we still conducted a new experiment on CrowdHuman [1] dataset, which is also a challenging datasets for dense pedestrian detection in the wild. The results listed below show that QFree-Det obtains overall higher performance compared to DINO variants, further confirming the effectiveness of our approach. We will include the experiment results on the CrowdHuman dataset in the new version of the paper.
>
> | Method | Epochs | AP$\uparrow$ |  Recall$\uparrow$ | mMR$\downarrow$ |
> | :---: | :---: | :---: | :---: | :---: |
> | DINO-ResNet50 | 24 | 86.61 | 95.11 | 52.85 |
> | **QFree-Det-ResNet50 (ours)** | 24 | **86.87** |  **95.25**  | **52.08** |
> | DINO-Strip-MLP-T | 24 | **88.38** | 95.82 | 50.30 |
> | **QFree-Det-Strip-MLP-T (ours)** |  24 | 87.92 | **95.99** | **49.83** |
> |  DINO-Swin-T | 24 | 87.71 | 95.71 | 51.81 |
> | **QFree-Det-Swin-T (ours)** | 24 | **88.14** |  **95.86** |  **51.12** |
> | DINO-VMamba-T | 24 | 87.44 | 95.45 | 51.54 |
> | **QFree-Det-VMamba-T (ours)** | 24 | **88.46** | **96.20** | **50.28** |
>
> **Q5.2 Specific scenarios.** The variation in detection scenarios reflects differing levels of detection difficulty, which significantly impacts model performance. The primary significance of QFree-Det lies in *its ability to transform a fixed-query detector into a free-query detector* while maintaining the model accuracy and not increasing computational requirements. Similar to many advanced models (such as Co-DETR and DAC-DETR), QFree-Det is designed to enhance model accuracy and improve detection robustness, but it may still faces challenges on blurriness and occlusion object detection. *These detection scenarios remain ongoing challenges to be addressed.*
>
> On the other hand, the accuracy of detection models is significantly influenced by the backbone used to extract *initial feature representations*, as this backbone forms the foundation for the model's subsequent decoder during detection. The variations in accuracy across different backbones can be quite substantial. For instance, QFree-Det-VMamba-T achieved improvements of +3.9\% AP and +4.5\% AP$_S$ over QFree-Det-ResNet50 (54.4\% vs. 50.5\% AP and 34.3\% vs. 38.8\% AP$_S$) under same 12 epochs training. We hope our method could spark new insights for developing high-accuracy, robust, query-free detection models in further research within the vision community.
>
>
> [1] Shao S, Zhao Z, Li B, et al. Crowdhuman: A benchmark for detecting human in a crowd[J]. arXiv 2018.

---

> ### Author Response · Authors · 2024-11-20
> **Responses to the Reviewer TZb2 (part 4)**
>
> **Q6: Detailed analysis of the trade-offs between performance and computational cost in QFree-Det.**
>
> It's a significant issue for the model design. For the model with an adaptive free number of queries, computational complexity and performance are related to the number of objects in the test images. In the Appendix **Sec.A.5** of the main paper, we have conducted ablation studies on the classification score $S$ using the COCO dataset, exploring how the number of queries and FLOPs change with variations in the classification score ***S***. The results are shown in **Table 11** of Appendix in the main paper. For your convenience, we have copied it below:
>
>
> | $S$ | AP | AP$_{50}$ | AP$_{75}$ | AP$_S$ | AP$_{M}$ | AP$_{L}$ |
> | :---: | :---: | :---: | :---: | :---: | :---: | :---: |
> | 0.01 | **50.6** | **67.9** | **55.4** | 34.1 | 54.3 | 64.8 |
> | 0.02 | 50.5 | 67.5 | 55.1 | **34.3** | **54.6** | 64.5 |
> | 0.03 | 50.2 | 67.4 | 54.6 | 33.2 | 53.7 | 64.8 |
> | 0.04 | 50.2 | 67.2 | 54.5 | 33.6 | 53.8 | 64.5 |
> | 0.05 | 50.3 | 67.5 | 54.8 | 33.4 | 54.0 | **65.0** |
> | 0.06 | 50.1 | 67.3 | 54.6 | 33.2 | 53.7 | 64.5 |
> | 0.07 | 49.8 | 66.7 | 54.5 | 32.8 | 53.4 | 64.1 |
> | 0.08 | 50.0 | 67.0 | 54.5 | 33.2 | 53.9 | 64.0 |
> | 0.09 | 49.6 | 66.7 | 54.0 | 32.1 | 53.3 | 63.8 |
> | 0.10 | 49.5 | 66.4 | 54.0 | 33.2 | 53.0 | 64.0 |
>
> From this table and **Fig.4** of main paper, we can observe that a lower threshold *S* leads to a higher number of queries, which improves the model's accuracy but also increases its computational complexity. When *S* ranges from 0.01 to 0.05, the model's performance varies in a small range of 50.2\% $\sim$ 50.6\% AP, indicating good robustness. To balance the trade-offs between performance and computational cost, we set $S=0.02$ for the configuration used in other experiments in the main paper. With this configuration, our model achieves higher performance (50.5\% vs. 49.0\% in AP) while maintaining smaller computational complexity (273G vs. 279G) and faster decoder inference speed (9.2ms vs. 14.1ms) than DINO.
>
> ---
>
> **Q7: How does the AFQS algorithm scale with increasing image complexity and object density?**
>
> Thanks for your valuable comment. In the **Sec.B.2** (line 849 to 885) of Appendix in main paper, we have tested the proportional relation between the number of objects and the number of queries. The results show that as the number of objects in an image increases, the rate at which the number of queries increases slows down. To better illustrate this, we calculated the ratio of the model's queries to the number of objects in the image, presented in the table (*row 5*) below:
>
>
> | Object Numbers | 1-5 | 6-10 | 11-15 | 16-20 | 21-25 | 26-30 | 31-35 | 36-40 | 41-45 | 46+ |
> | :---: | :---: | :---: | :---: | :---: | :---: | :---: | :---: | :---: | :---: | :---: |
> | Image Numbers | 2769 | 985 | 556 | 324 | 170 | 83 | 29 | 22 | 9 | 5 |
> | Average Objects | 2.60 | 7.70 | 13.02 | 17.69 | 22.74 | 27.35 | 33.31 | 37.68 | 42.77 | 55.00 |
> | Query Numbers | 65.22 | 180.46 | 312.47 | 443.98 | 537.91 | 645.16 | 699.82 | 812.41 | 872.00 | 900.00 |
> | Query Num/Object Num | 25.08 | 23.44 | 23.99 | 25.10 | 23.65 | 23.59 | 21.01 | 21.56 | 20.39 | 16.37 |
> | DINO | 60.6 | 51.0 | 46.9 | 44.0 | 40.1 | 39.0 | 40.9 | 39.3 | 37.1 | 36.0 |
> | **QFree-Det (ours)** | **61.5** | **51.8** | **48.0** | **44.9** |  **40.1**|  **39.0** | **42.3** | **41.5** | **39.1** | **39.3** |
>
> From the table, it can be observed that as the number of objects increases, the growth rate of queries *slows down* and gradually declines, falling from 25.08 to 16.37. Concurrently, our model demonstrates a *growing advantage* over DINO as the number of objects increases across the subsets (31–35, 36–40, 41–45, 46+), as illustrated in **Fig.5** of Appendix in main paper. Notably, the subset of 1-5 objects accounts for 55.9\% of the images in the val2017 set of COCO. For this subset, our model utilizes only **7.25\%** of the queries (65.22 vs. 900) while achieving a higher performance with a **+0.9\%** AP, suggesting that the queries selected via AFQS are more effective.
>
> In terms of complexity, we have discussed the changes in FLOPs with varying queries in Appendix **A.5** of the main paper. As shown in **Fig.4**, the reduction in the number of queries (cyan dashed line) effectively lowers the model's FLOPs (purple solid line). These experiments underscore the effectiveness of our method's adaptive characteristics in object detection.

---

### Official Review · Reviewer_Pjik · 2024-11-07

**Soundness:** 3
**Presentation:** 3
**Contribution:** 2
**Rating:** 5
**Confidence:** 5

**Summary:**

The paper introduces a novel query-free detector, QFree-Det, aimed at addressing the limitation of a fixed number of object queries in Transformer-based detectors such as DETR and DINO. The authors propose an Adaptive Free Query Selection (AFQS) algorithm and a sequential matching method, which significantly enhance the model's capability to dynamically detect a variable number of objects across different input images. Additionally, a new loss function, PoCoo, is designed to improve detection capabilities. The experimental results demonstrate that QFree-Det achieves remarkable performance on multiple datasets and various backbone networks.

**Strengths:**

> 1. Motivation:

The paper introduces the QFree-Det model, addressing the variable number of instances in different images, which is a pertinent issue in object detection. The proposal for a dynamic query selection approach is innovative and could significantly enhance the flexibility and accuracy of detection models.

> 2. Performance Improvement:

The paper demonstrates some performance improvements on the WiderPerson dataset, particularly in detecting small objects, which is a challenging aspect of computer vision.

**Weaknesses:**

> Comparison with DETA and DDQ-DETR:

The discussions in Section 3.1.3 regarding the role of Self-Attention (SA) bear similarities to the disscussions taken in "NMS Strikes Back" (DETA) [1] and "Dense Distinct Query for End-to-End Object Detection" (DDQ-DETR) [2]. Both papers explore the effectiveness of traditional non-maximum suppression (NMS) and the role of SA integrated with transformer-based detectors, which is a relevant area of comparison for QFree-Det.

[1] https://arxiv.org/pdf/2212.06137

[2] Dense Distinct Query for End-to-End Object Detection (CVPR2023)

> Comparative Experiments in Table 3:

While the paper provides comparisons with models like DINO, it lacks a comprehensive set of comparative experiments with other state-of-the-art Transformer detectors, such as Stable DINO and CO-DINO. Including these comparisons would provide a more holistic view of QFree-Det's performance relative to the current landscape of object detection models.

> Performance Comparison with Fixed Detectors:

The paper compares the performance of QFree-Det (AP 50.5) with DINO (AP 49.0). However, it is noted that improved versions of DINO, utilizing focal loss, have achieved an AP of 50.0, as demonstrated in the MMDETECTION repository. This raises the question of whether other state-of-the-art fixed detectors, when combined with loss functions like PoCoo or IA-BCE, could potentially outperform QFree-Det. Further experimentation and comparison with these models would be beneficial to assert the superiority or uniqueness of QFree-Det's approach.

**Questions:**

> The number of decoder layers.

It seems decoder only have one layer of CA and 6 layers of SA?

> Speed of inference.

How fast can the QFree-Det be for some sparse scenario, i.e., only a instance in a image.

> Performances in line 476

Is the same experiments (47.1) as that of line 248?

---

> ### Author Response · Authors · 2024-11-20
> **Responses to the Reviewer Pjik (part 1)**
>
> Dear reviewer,
>
> We sincerely thank you for putting in the time to provide us with a thorough review. We have carefully considered your suggestions and have done our best to address these concerns.
>
>
> **Q1: Comparison with DETA and DDQ-DETR: the disscussions about NMS and Self-attention (SA).**
>
> Thanks very much for the valuable comments and suggestions.
>
> DETA [1] and DDQ-DETR [2] both use NMS to eliminate duplicates in their models. Removing duplicate detection boxes is a crucial step for reducing false positive samples in detection systems. We categorize existing methods of deduplication into two types based on their principles: ***box-based*** and ***class-based***. Box-based methods, such as NMS, work by *comparing the overlapping (IoU) between predicted boxes*. While straightforward, they are sensitive to the IoU threshold and struggle with overlapping targets. In contrast, class-based methods utilize self-attention to compare features between queries, influencing classification scores to achieve deduplication, which results in *lower scores for redundant boxes*. This approach can mitigate the issue of overlapping objects and is commonly employed in DETR-like models. Intuitively, combining both *box-based* and *class-based* methods may further enhance the model's overall performance.
>
> Compared to DETA and DDQ-DETR, our approach **addresses different problems** (in motivation and objective), **introduces different solutions to issues** (in query selection, decoder architecture, and loss function), and **achieves comparable results** in performance. In addition, *these differences do not diminish the contributions of each method to solving the respective issues.*
>
> DETA is a *box-based* method for deduplication. Specifically, DETA is designed to investigate the impact of *one-to-many assignment-based training* on enhancing the training efficiency for DETR models, which **differs** to our motivation and objective. It achieves this by employing one-to-many IoU assignments in conjunction with NMS method, which is applied during both query selection and final prediction post-processing. This study shows that the *one-to-many **IoU assignments**, combined with NMS*, effectively improve training efficiency.
>
> DDQ-DETR combines both *box-based* and *class-based* deduplication methods. It specifically explores how *query distinctness* affects the model's optimization process and accuracy, which also **differs** to our motivation and objective. DDQ-DETR introduces the Distinct Query Selection (DQS) module, which uses training-unaware NMS to filter dense queries into distinct queries (*box-based method*). Then, DDQ-DETR further applies Hungarian matching (*class-based method*), considering both bounding box scores and class scores, to generate final one-to-one detection results. Additionally, DDQ-DETR also uses one-to-many label assignments and incorporates an *auxiliary head* along with Auxiliary Loss for Dense Queries to maintain training efficiency (but this kind of mixing label assignments on same decoder weights still introduces the issue of ''*detection ambiguity*''). Overall, this approach highlights that both sparse and dense queries in end-to-end detection are problematic. By explicitly combining box-based and class-based methods, DDQ-DETR ultimately enhances model accuracy.
>
> Unlike these two methods, our approach focuses on transforming a *fixed-query* detector into a *free-query* detector, thereby addressing the fixed capacity of DETR-like models. We deeply explore the role of SA (which has NOT been examined in DDQ-DETR) to address challenges related to the existing decoder structure during this transition, particularly the training GPU memory demands issue associated with SA. In contrast to DETA and DDQ-DETR, **our object is not to investigate how to enhance model accuracy through NMS**. Furthermore, we observe that the existing connection between CA and SA can result in the ``recurrent shifting`` problem (as noted in *line 273* of the main paper). To tackle these challenges, we developed a more effective decoder structure (LDD) and implemented *decoupled* one-to-one and one-to-many label assignments. This design significantly alleviates the ``detection ambiguity`` issue (*Table 12 and Table 16* in Appendix of the main paper). Notably, DETA and DDQ-DETR share a similar decoder structure with existing methods, which highlights the contribution of our approach of novel LDD decoder. Compared to previous class-based deduplication method, QFree-Det *further optimizes* the decoder structure, enabling *query-free* detection while minimizing the ''*detection ambiguity*" associated with one-to-many assignments.

---

> > ### Author Response · Authors · 2024-11-20
> > **Responses to the Reviewer Pjik (part 3)**
> >
> > **Q3: Performance Comparison with Fixed Detectors: other SOTA fixed detectors with PoCoo or IA-BCE loss could potentially outperform QFree-Det?**
> >
> > Thank you for your valuable feedback. This is an excellent question that merits further exploration. The loss function is a general approach that can be applied to different models. Intuitively, it can be used across various models to further enhance accuracy.
> >
> > Actually, there are numerous factors influencing model accuracy, such as codebase of model frameworks, different data-augmentation, hyper-parameter settings, loss functions, and etc.
> > However, our model is not built on the foundation of latest SOTA models and codebase, nor is it intended to surpass the performance of all existing SOTA models. Instead, our objective is to develop a ``free-query`` detector and address the ``detection ambiguity`` issue present in existing model decoders.
> >
> > We validated the *general* effectiveness of each module in our model through ablation experiments conducted on the baseline DINO model across **five** different backbone models (in *Table 3, Table 5 and other extensive experiments* in Appendix of the main paper). However, as suggested, we carried out *additional* experiments on PoCoo loss. Due to the time and GPU resource constraints, we focused our experiments on the Stable-DINO model (also mentioned by the reviewer in **Q2**). We directly added the PoCoo loss to the loss function of Stable-DINO, noted as Stable-DINO-PoCoo. The results are showen as follows:
> >
> > | Model | Backbone | Objects | Codebase | Epochs | AP | AP$_{50}$ | AP$_{75}$ | AP$_S$ | AP$_{M}$ | AP$_{L}$ |
> > | :---: | :---: | :---: | :---: | :---: | :---: | :---: | :---: | :---: | :---: | :---: |
> > | Stable-DINO | ResNet50 | fixed | detrex | 12 | 50.4 | 67.4 | 55.0 | 32.9 | 54.0 | **65.5** |
> > | Stable-DINO-PoCoo | ResNet50 | fixed | detrex | 12 | 50.5 | 67.2 | 55.1 | 34.1 | 54.2 | 64.7 |
> > | **QFree-Det**  | ResNet50 | *free* | DINO-official-public | 12 | 50.5 | **67.5** | 55.1 | 34.3 | **54.6** | 64.5 |
> > | **QFree-Det-PMM** | ResNet50 | *free* | DINO-official-public | 12 | **50.7** | 67.4 | **55.5** | **34.4** | 54.4 | 64.9 |
> >
> > The experimental results show that our PoCoo loss further enhances the accuracy of Stable-DINO by **+0.1%** AP and **+1.2%** AP$_S$, making it comparable to our QFree-Det model. Moreover, for a fair comparison, we developed a new variant of *QFree-Det-PMM* by applying the Position-Modulated Matching (PMM) approach from Stable-DINO into our QFree-Det model, resulting in a slight performance improvement that surpasses Stable-DINO and reaffirms the effectiveness of our method.
> >
> >
> >
> > In addition, as illustrated in **Fig.3** and **Table 8** of the main paper, when our method is directly applied to other transformer-based detectors like Deformable DETR [2], the new variant, Deformable DETR-QFree, outperforms the original Deformable DETR by **3.1\%** AP (49.26\% vs. 46.20\%) and converges faster. This improvement occurs even **without using the denoising training and the look-forward twice approach employed in DINO**, further demonstrating the effectiveness and generalizability of our approach.
> >
> >
> > [1] Liu S, Ren T, Chen J, et al. Detection transformer with stable matching[C]//ICCV 2023.
> >
> > [2] Zhu X, Su W, Lu L, et al. Deformable DETR: Deformable Transformers for End-to-End Object Detection[C]//ICLR 2021.
> >
> > ---
> >
> > **Q4: The number of decoder layers: It seems decoder only have one layer of CA and 6 layers of SA?**
> >
> > As shown in *Fig.2* and *Table 12* of the main paper, our LDD decoder architecture consists of 4 BLP layers (with 4 CA modules) and 2 DP layers (containing 2 CA and 4 SA), for a total of **6 CA and 4 SA** modules. In contrast, existing DETR-like models include 6 decoding layers, comprising **6 CA and 6 SA** modules, making our LDD framework more efficient.
> >
> > In addition, compared to other one-to-many methods such as H-DETR, DAC-DETR, Co-DETR, our LDD architecture *does NOT require extra decoder branches or multiple auxiliary heads*.
> > This makes our LDD decoder framework more streamlined and efficient, effectively utilizing the roles of CA and SA while appropriately allocating one-to-one and one-to-many label assignments in different decoder layers, addressing the detection ambiguity issue. As a result, our model offers advantages in terms of lower complexity (see Appendix **A.5** of the main paper), higher accuracy in both sparse and dense scenes (see Appendix **B.2** of the main paper), and improved decoder inference speed, achieving a **34.8\%** improvement over DINO (as noted in the **General Response**).

---

> ### Author Response · Authors · 2024-11-20
> **Responses to the Reviewer Pjik (part 2)**
>
> **Continue the answer to Q1:**
>
> The experimental results are presented in the table below:
>
> | Model | Backbone | Objects | Codebase | Epochs | AP | AP$_{50}$ | AP$_{75}$ | AP$_S$ | AP$_{M}$ | AP$_{L}$ | Params |
> | :---: | :---: | :---: | :---: | :---: | :---: | :---: | :---: | :---: | :---: | :---: | :---: |
> | DETA$^*$ | ResNet50 | fixed | Detectron2 | 12 | 50.5 | 67.6 | 55.3 | 33.1 | 54.7 | 65.2 | 52M |
> | DDQ-DETR | ResNet50 | fixed | MMDetection V3.0 | 12 | 51.3 | 68.6 | 56.4 | 33.5 | 54.9 | 65.9 | - |
> |**QFree-Det (ours)** | ResNet50 | *free* | DINO-official-public | 12 | 50.5 | 67.5 | 55.1 | 34.3 | 54.6 | 64.5 | 48M |
>
> (*Note: DETA with * uses 9 encoder layers. Detectron2 and MMDetection are both improved codebase for DETR-like detector.*)
>
> It is important to note that our model was developed using the public official DINO [3] code, rather than improved frameworks such as DETA with *Detectron2* [4] and DDQ-DETR with *MMDetection V3.0* [5]. This may lead to an unfair comparison of their accuracy, as the variations in codebases have not been eliminated. Despite this, our model still achieves comparable results and even outperforms others in small object detection, demonstrating its effectiveness. The suboptimal implementation of our model may not fully showcase its advantages, as the reviewer has mentioned in **Q3** regarding the improved *MMDetection* repository. Due to time constraints, we will leave this issue in future work.
>
> [1] Ouyang-Zhang J, Cho J H, Zhou X, et al. Nms strikes back[J]. arXiv preprint, 2022.
>
> [2] Zhang S, Wang X, Wang J, et al. Dense distinct query for end-to-end object detection[C]//CVPR 2023.
>
> [3] Zhang H, Li F, Liu S, et al. DINO: DETR with Improved DeNoising Anchor Boxes for End-to-End Object Detection[C]//ICLR 2022.
>
> [4] Wu Y, Kirillov A, et al. Detectron2. 2022.
>
> [5] Chen K, Wang J, Pang J, et al. MMDetection: Open MMLab Detection Toolbox and Benchmark. arXiv preprint 2019.
>
> ---
>
> **Q2: Comparative Experiments in Table 3: comparison with Stable-DINO and CO-DINO.**
>
> Thanks for the valuable comment. The comparison of Co-DETR [1] with our method was listed in Table 3 (line 389) of the main paper. Following the suggestions, we further list the results about Stable-DINO [2] method as below:
>
> | Model | Backbone | Objects | Codebase | Epochs | AP | AP$_{50}$ | AP$_{75}$ | AP$_S$ | AP$_{M}$ | AP$_{L}$ |
> | :---: | :---: | :---: | :---: | :---: | :---: | :---: | :---: | :---: | :---: | :---: |
> | Stable-DINO | ResNet50 | fixed | detrex [3] | 12 | 50.4 | 67.4 | 55.0 | 32.9 | 54.0 | **65.5** |
> | Co-DETR-4scale | ResNet50 | fixed | MMDetection | 12 | 49.5 | **67.6** | 54.3 | 32.4 | 52.7 | 63.7 |
> | **QFree-Det (ours)** | ResNet50 | *free* | DINO-official-public | 12 | **50.5** | 67.5 | **55.1** | **34.3** | **54.6** | 64.5 |
>
> (*Note: detrex and MMDetection are both improved codebase for DETR-like detector*.)
>
> As we can see, compared to these two methods, our approach achieves the overall best performance, especially for the small objects detection, increasing by +1.4\% and +1.9\% AP$_S$ than Stable-DINO and Co-DETR, respectively. We will include this comparison with the Stable-DINO model in Table 3 of the new version of the main paper.
>
> [1] Zong Z, Song G, Liu Y. Detrs with collaborative hybrid assignments training[C]//ICCV2023.
>
> [2] Liu S, Ren T, Chen J, et al. Detection transformer with stable matching[C]//ICCV2023.
>
> [3] Ren T, Liu S, Li F, et al. detrex: Benchmarking detection transformers[J]. arXiv preprint 2023.

---

> ### Author Response · Authors · 2024-11-20
> **Responses to the Reviewer Pjik (part 4)**
>
> **Q5: Speed of inference: How fast can the QFree-Det be for some sparse scenario, i.e., only a instance in a image?**
>
> Thank you for the comment. We conducted additional comprehensive tests on inference speed, varying the number of queries from 10 to 1800. The results indicate that, with the same number of queries as DINO (900 queries), our LDD decoder framework has improved the inference speed of the model's decoder by **34.8\%** over DINO. When the number of queries is further reduced, the decoder’s inference speed remains stable, primarily due to the parallel computation performed by CA and SA within the decoder. For more details, please refer to our responses in the **General Responses** section.
>
> On the other hand, our query-free model, which adapts the number of queries based on the image itself, holds significant potential for open-ended object detection task. This task directly generates object names through a generative large language model, where the query count greatly influences the model's complexity. The adaption characteristic of our model enables reduced queries to feed into generative large language model, thereby decreasing the workload for subsequent category generation processing, which highlights its potential applications and importance for the future research in vision community.
>
> ---
>
> **Q6: Is the same experiments (44.1) as that of line 248?**
>
> Yes, it is the same experiment in *line 476* and *line 248*. This experiment is conducted to test the impact on model accuracy after removing SA. By training the model without SA, we can perform multiple tests with any number of queries **without needing to retrain the model**. However, this model still lacks dynamic adaptability to queries and can only perform testing with manually defined varied number of queries.
>
> In line 248, we labeled it as ''*fixed query number*'' because the test is still conducted with a fixed query count (900). In line 476, we labeled it as ''*free*'' because this model can accommodate *any number of query tests*, without the need to retrain the model. It may not be appropriate to directly label it as ''free'' directly, so we will update it to ''*free-conditioned query*'', indicating that it is free but subject to the constraints of the manually defined testing parameters.

---

> > ### Comment · Reviewer_Pjik · 2024-11-20
> >
> > Thanks for the responses.
> >
> > Firstly I think the performance improvments are minior comparing to sota fixed detectors, i. e., 0.1~2 comparing to stable dino in Q2 and Q3, which weaken your contributions.  If this is a codebase-related problem, you should fix it then.
> >
> > Secondly, I mean the performances of CO-DINO which can achieve 52.1AP instead of CO-DETR in Q2. But I think you do not have to compare it directly to CO-DINO as they used multiple branches in training. You can actually add your branch on CO-DETR to replace the DINO branch, then you might get higher performances. But this is not essential to your contribution, you can do it later. This will not influence my score.
> >
> > Now, my primary concern is the advantages of query-free detectors over query-fixed detectors. Are they inherently stronger, more robust, faster, or theoretically more elegant? Additionally, the critique provided by Reviewer eExB is quite incisive, offering a sharp perspective on this matter. Anyway, I do think your motivation is cool, adpative query number could be useful in some traffic scenarios, e.g., one batch has several cars, while other one may have thousands of cars.

---

> > > ### Author Response · Authors · 2024-11-20
> > >
> > > Dear reviewer,
> > >
> > > We really appreciate your quick response. While Co-DINO [1] achieves an accuracy of **52.1 AP**, it is highly unfair to directly compare it with a **4-scale** detector, such as our QFree-Det model. Co-DINO utilizes a **5-scale** backbone features with a total computational cost exceeding ``860G``, which is approximately **three times compared to 4-scale models**, like DINO with 279G and our model with **275G** (in 900 queries).
> > >
> > > Additionally, the 5-scale model also requires significantly more memory and cost during training,  as noted in the GitHub issues of the official Co-DINO code [1]. Currently, there are few comparisons of 5-scale models in existing transformer-based detectors.
> > > The accuracy improvement of the 5-scale model over the 4-scale model is **not cost-effective**. Under **12** epochs of training, our QFree-Det-VMamba-T-4scale achieved an accuracy of **54.4%** AP (Table 3 of the main paper), which is significantly higher than the 52.1% AP. I also attempted to find results for Co-DINO-4scale-ResNet50 in other papers or on public websites, but unfortunately, there is no publicly available information, including official code.
> > >
> > > | Model | Backbone | Objects | Codebase | Epochs | AP | AP$_{50}$ | AP$_{75}$ | AP$_S$ | AP$_{M}$ | AP$_{L}$ |
> > > | :---: | :---: | :---: | :---: | :---: | :---: | :---: | :---: | :---: | :---: | :---: |
> > > | Co-DETR-4scale | ResNet50 | fixed | MMDetection | 12 | 49.5 | 67.6 | 54.3 | 32.4 | 52.7 | 63.7 |
> > > | Co-DINO-5scale | ResNet50 | fixed | MMDetection | 12 | **52.1** | 69.4 | 57.1 | 35.4 | 55.4 | 65.9 |
> > > | **QFree-Det-4scale (ours)** | ResNet50 | *free* | DINO-official-public | 12 | 50.5 | 67.5 | 55.1 | 34.3 | 54.6 | 64.5 |
> > >
> > > [1] Zong Z, Song G, Liu Y. Detrs with collaborative hybrid assignments training[C]//ICCV2023.

---

> > > > ### Comment · Reviewer_Pjik · 2024-11-20
> > > >
> > > > Thanks for the responses. This totally solved my question about the CO-DINO. Then the only problem left is the advantages of query-free detectors over query-fixed detectors, now it looks the difference between two paradiams is minor.

---

> > > > > ### Author Response · Authors · 2024-11-21
> > > > > **Responses to the Reviewer Pjik (1/2)**
> > > > >
> > > > > Dear reviewer,
> > > > >
> > > > > Thank you again for your quick response. We are very pleased to have effectively addressed some of your concerns.
> > > > >
> > > > > The primary limitation of *query-fixed* detectors is their requirement to predict a **large** and **fixed** number of detection results for both **sparse** and **dense** scenes. On one hand, this results in a considerable amount of *unnecessary and redundant computation*. On the other hand, when these fixed-query detectors are applied to more challenging downstream tasks, such as Open-Ended Detection [1] (which is a multi-modal task), these redundant queries would be further fed into the **large language model** (LLM) to generate object names directly, which will significantly increase the overall computational complexity and cost.
> > > > >
> > > > > To demonstrate this, we conducted a simple test using the public official code of GenerateU [1]. We evaluated the computational complexity of the generative large language model used in GenerateU. With 900 queries of the detector fed into the LLM, the complexity is **10,139.64 GFLOPs**. In contrast, with 200 queries, the complexity siginificantly dropped to **2,253.45 GFLOPs**, which indicates that reducing the number of queries can significantly decrease the computational load of the LLM model for the open-ended detection multi-modal task. We will include the analysis of the influence of query-free model on downstream tasks in the new version of the paper.
> > > > >
> > > > >
> > > > > In contrast to fixed-detectors, our *query-free* detector *adaptively* eliminate redundant queries at *an early stage* in the transformer-based detector. This leads to a **more efficient, cost-effective, accurate, and flexible approach**, as demonstrated in the following four aspects:
> > > > >
> > > > > (1) **High-rate of effective query utilization, better cost-efficiency, and higher performance.** In *sparse scenarios*, QFree-Det achieves comparable or even higher accuracy with only a small number of queries, while simultaneously reducing computational load of the decoder. The transformer-decoder is primarily composed of layers of cross-attention (CA) and self-attention (SA). CA has a complexity of **$o(NKC^2)$** (using deformable attention [2], where $N$ is the number of queries, K is the number of sample points, and $C$ denotes the number of channels), while SA has a complexity of **$o(N^2)$**. As the number of queries $N$ decreases, the computational load of the CA and SA reduces at a *linear* and *quadratic* rate, respectively. This makes the approach *highly cost-effective* and leads to *lower power consumption* during deployment, particularly in *edge devices*, which has significant practical applications.
> > > > >
> > > > > In many scenarios, objects within the image are often sparse. For instance, in the val2017 of COCO dataset, **55\%** of images contain between 1 and 5 objects (as shown in Appendix **B.2**, **Table 9** of the main paper). In this context, we achieved an accuracy that is **+0.9\%** AP higher than the DINO model while using only **7.25\%** of the queries (65.22 vs. 900), clearly demonstrating the effectiveness of our query with high-rate utilization.
> > > > >
> > > > > (2) In dense scenarios, our query-free model also achieves higher accuracy with fewer queries than DINO. This robust advantage becomes even more pronounced as the number of objects increases, as illustrated in **Fig.5** and **Table 9** of the main paper (specifically in the subsets of $31\sim35$, $36\sim40$, $41\sim45$, $46+$).
> > > > >
> > > > > (3) For more complex *multi-modal* detection tasks, such as open-ended detection [1], decoder queries are input into **large language models** to directly generate corresponding object names without additional vocabulary priors. This process is highly **complexity-sensitive** to the number of *queries* (for instance, the LLM model shows 10,139.64 GFLOPs with 900 queries and 2,253.45 GFLOPs with 200 queries [1]). However, fixed-query approaches utlize a large fixed number of queries, which would significantly increase the computational load. In contrast, our query-free method reduces this redundancy by eliminating unnecessary queries at an early stage, which is highly significant and holds great potential for these multi-modal tasks.
> > > > >
> > > > > (4) Our query-free method offers greater **flexibility** through its inherent adaptive characteristics. For example, once the model is trained, the number of queries can be adaptively adjusted to suit *various scenarios* without the need for retraining, which is *still required for fixed-query detectors*.
> > > > >
> > > > >
> > > > > [1] Lin C, Jiang Y, Qu L, et al. Generative Region-Language Pretraining for Open-Ended Object Detection[C]//CVPR 2024.

---

> > > > > ### Author Response · Authors · 2024-11-21
> > > > > **Responses to the Reviewer Pjik (2/2)**
> > > > >
> > > > > **Continue the responses:**
> > > > >
> > > > > Furthermore, our model has demonstrated superior performance on the COCO dataset (Table 3 of the main paper), as well as on the more challenging WiderPerson (Table 4 of the main paper) and CrowdHuman (responses to reviewer TZb2 of Q5.1) datasets than the DINO model. This further indicates its **robustness** across a variety of scenarios.
> > > > >
> > > > > In addition, we also introduce a novel decoding framework, LDD, which effectively tackles the issue of ''*detecting ambiguity*'' caused by mixing label assignments of one-to-one and one-to-many, as well as the ''*recurrent shifting*" problem. The effectiveness of our decoding framework is demonstrated by the experimental results in **Table 12** and **Table 16** of the main paper. It *enhances* performance while **simplifying** the decoder architecture by using *fewer* SA layers, a single adaptive query type, and *eliminating the need for additional branches or multi-head prediction modules*.
> > > > >
> > > > > Finally, our LDD framework significantly improves the speed of the decoder by **+34.8\%** compared to DINO (as detailed in **General Responses**). Although the overall speed advantage of the detector may not be substantial due to the multiple components of transformer-based detectors—where their slow inference speed are a common bottleneck, especially when compared to faster CNN-based detectors like YOLO [3])— our approach with improved speed of decoder offers a promising solution for enhancing the speed of transformer-based detectors. For instance, by integrating the YOLO *backbone*, the Sparse-DETR [4] *encoder*, and our LDD *decoder*, our approach shows great potential to develop a *high-speed, low-cost, high-performance, query-free* transformer detector, which deserves further exploration in future work.
> > > > >
> > > > >
> > > > > [1] Lin C, Jiang Y, Qu L, et al. Generative Region-Language Pretraining for Open-Ended Object Detection[C]//CVPR 2024.
> > > > >
> > > > > [2] Zhu X, Su W, Lu L, et al. Deformable DETR: Deformable Transformers for End-to-End Object Detection[C]//ICLR 2021.
> > > > >
> > > > > [3] Khanam R, Hussain M. YOLOv11: An Overview of the Key Architectural Enhancements[J]. arXiv preprint 2024.
> > > > >
> > > > > [4] Roh B, Shin J W, Shin W, et al. Sparse DETR: Efficient End-to-End Object Detection with Learnable Sparsity[C]//ICLR 2022.
> > > > >
> > > > > Kind regards,
> > > > >
> > > > > Authors of Paper ID: 4457

---

> > > > > ### Author Response · Authors · 2024-12-03
> > > > > **Kindly remainder for the feedback in the final discussion phase.**
> > > > >
> > > > > Dear Reviewer Pjik,
> > > > >
> > > > > We deeply appreaciate your valuable time and efforts in reviewing our paper. We hope that our new responses (1/2 & 2/2), along with the carefully revised manuscript, address your remained concerns. If all issues have been resolved, we would be grateful if you could kindly reconsider your rating based on the revised manuscript and all clarifications. We highly value your constructive and thorough reviews, and we are committed to improving our paper to meet your expectations.
> > > > >
> > > > > Thank you once again for your time and consideration.
> > > > >
> > > > > Best regards,
> > > > >
> > > > > Authors

---

### Author Response · Authors · 2024-11-20
**General Responses About: inference speed tests and analysis**

Dear Reviewers,

We sincerely thank you for your thorough and valuable reviews. We believe your suggestions have improved our paper. In this section, we present the General Responses about the *inference speed tests and analysis*.

In **Sec.A.5** and **Sec.B.2** of the Appendix in the main paper, we have conducted experiments on COCO and WiderPerson, along with analyses to explore the relationship between the adaptive number of queries, computational complexity, and model accuracy. The results highlight the advantages of our method in both aspects.

Following suggestions, we conduct additional tests on the inference speed of QFree-Det. For a fair comparison of our model with the baseline model DINO, the test was applied on the same codebase (official published code of DINO), backbone model (ResNet50), input image size (1280 $\times$ 800), GPU device (RTX 3090), PyTorch lib (pytorch 2.2), and CUDA version (Driver Version: 535.183.01, CUDA Version: 12.2). We tested the DINO (900 query) model and QFreeDet (query varies from *10 to 1800*) model's inference speed, and the results are shown as below:


| Model | DINO | QFree-Det | QFree-Det | QFree-Det | QFree-Det | QFree-Det |
| --- | --- | --- | --- | --- | --- | --- |
| Query Number |  900 | 1800 | 900 | 500 | 100 | 10 |
| Backbone (ms) | 15.2 | 15.2 | 15.2 | 15.2 | 15.2 | 15.2 |
| Encoder (ms) |  30.6 | 30.6 | 30.6 | 30.6 | 30.6 | 30.6 |
| Query Selection (ms) | 7.3 | 7.8 | 7.5 | 7.4 | 7.4 | 7.4 |
| Decoder (ms) | 14.1 | 11.0 | 9.2 | 8.9 | 8.9 | 8.8 |
| Inference Time (ms) | 67.2 | 64.6 | 62.5 | 62.1 | 62.1 | 62.0 |
| FPS (frame/s) | 14.9 | 15.5 | 16.0 | 16.1 | 16.1 | 16.1 |

The inference time of both models is composed of four components: *backbone time, encoder time, query selection time, and decoder time*. Since the backbone and encoder components are identical in both models, their inference speeds are also the same. The inference time of query selection process in both models is similar, with most of the duration spent on the model classification head in predicting scores (about 7ms) among all encoder tokens for subsequent selection. For our AFQS algorithm, it then retrieves queries based on these scores within **1 ms**, which is faster in query selection process of transforming the fixed-query into a free-query for the DETR detector.

Additionally, we observed that under the same query conditions, such as using *900 queries*, our LDD framework significantly enhances the inference speed of decoder, achieving a **+34.8\%** improvement compared to DINO. This result clearly validates the effectiveness of our designed novel LDD decoder framework. When the number of queries is further reduced, the decoder's inference speed remains stable, primarily due to the parallel computation performed by CA and SA in the decoder.

It is noted that to improve the model's inference speed is not the primary objective of our method. The above table indicates that the backbone and encoder models account for **73\%** of the inference time. This observation provides insights for further optimizing these components to accelerate the inference speed of DETR models. Notably, our efficient LDD decoder framework has successfully increased the inference speed of the model's decoder by **34.8\%**. Thus, this framework can be integrated with other model architectures, such as the YOLO series, not only to further enhance overall inference speed but also achieve the detection of a free number of objects.

Additionally, this query-free model, which adapts the number of queries based on the image itself, *holds significant potential for open-ended object detection task*. This task directly generates object names through a generative **large language model**, where the query count greatly influences the model's complexity. The *adaption characteristic* of our model enables reduced queries to feed into generative large language model, thereby decreasing the workload for subsequent category generation processing, which highlights its potential applications and importance for the future research in vision community.

Kind regards,

Authors of Paper ID: 4457

---

### Author Response · Authors · 2024-11-25
**Revision**

Dear reviewers,

We sincerely appreciate your thorough and detailed reviews. We believe your suggestions have improved our paper. In the following, we present the changes made in the revised version, which are highlighted in blue. We have also provided individual responses to each of your concerns under respective reviews. If you have any additional comments or suggestions, we would be grateful for the opportunity to clarify further.

- We have updated the query type in Table 5 from ''free'' to ''free-conditioned query''.
- We have added comparisons of Stable-Dino, DDQ-DETR, DETA, MS-DETR in Table 3 to enhance the comparison with SOTA models. Additionally, we included experiments on CrowdHuman in Table 19 of the Appendix to further validate the effectiveness of our method.
 - We have added the ratio of query numbers to object numbers in Table 9 (row 5) of the Appendix to further illustrate the relation between the number of objects and queries, highlighting the advantages of the query-free method.
 - We have added a new Fig.6 (visualization of row 2 and row 4 of Table 9) to illustrate the trend between the number of queries selected by QFree-Det model and the number of objects.
 - We have added a new section titled ''*Inference Speed Tests and Analysis*'' (Sec.D) in the Appendix, which includes the ''Speed Tests'' and the ''Potential Effectiveness on Downstream Open-Ended Detection Tasks'' to show the potential applications of the query-free model in downstream tasks.
 - We have added a new ``Discussion`` section (Sec.E) in the Appendix, which primarily explores the differences and effectiveness of our method compared to existing detection methods, along with the advantages and potential applications of our QFree-Det model. This section includes the following topics:

 (1) Sec.E.1: ''The Deduplication Roles of Self-Attention (SA) and NMS'' and ''Deduplication Part (DP) vs. NMS'';

 (2) Sec.E.2:  ''The Differences of Sequential Matching with Existing Methods'' and ''Differences between H-DETR, MS-DETR and QFree-Det on Sequential Matching'';

 (3) Sec.E.3: ''The impact of removing PQ'' and ''LDD framework for addressing Matching Ambiguity'';

 (4) Sec.E.4: ''The Computational Complexity Analysis'';

 (5) Sec.E.5: ''The Contributions of AFQS Algorithm'' ;

 (6) Sec.E.6: ''Advantages of the QFree-Det Model'';


Kind regards,

Authors of Paper ID: 4457

---

### Author Response · Authors · 2024-11-28
**Common Response**

Dear reviewers and ACs,

We sincerely appreciate your valuable time, diligent efforts, and thorough reviews.

We propose QFree-Det, a novel query-free model that achieves ''*free-object predictions*'' through the AFQS algorithm and a more efficient LDD architecture. This model effectively addresses the limitations of ''*fixed-query*'' and ''*detection ambiguity*'', achieving high-rate of effective query utilization, improved cost-efficiency, and enhanced performance. Additionally, it strengthens the detector's adaptive detection capabilities, accelerates the inference speed of the decoder, and shows promising potential for downstream open-ended detection tasks.

We are delighted to note that reviewers find that:
* Cool motivation (``Pjik``);
* Innovative/novel/interesting solution/idea for achieving free-query detection (``Pjik``, ``TZb2``, ``eExB``); Novel and the first method for addressing the ambiguity problem (``rMNn``);
* Demonstrated improved performance and effectiveness (``Pjik``, ``TZb2``), robust framework for diverse detection scenarios (``TZb2``), enhancing for small objects detection (``Pjik``, ``TZb2``), the comprehensive experiments (``rMNn``), good performance compared to SOTA (``TZb2``, ``eExB``);
* Well-written/good presentation/easy to follow in writing style (``rMNn``, ``eExB``).

We deeply appreciate your constructive feedback on our manuscript. Your insights are highly valued, and we believe your suggestions have improved our paper. We have carefully revised the manuscript and addressed each of your concerns with additional tests, experiments, and detailed analyses. We hope our individual responses (accompanying the revised PDF) have carefully addressed each of your concerns. We are respectfully request that the revieweres to kindly reconsider the ratings based on our responses and the revised manuscript. If you have any additional comments or suggestions during the discussion period, please feel free to reach out, and we would greatly appreciate to provide further clarification.

Thank you once again for your time and consideration.

Best Regards!

Authors

---

### Meta-Review · Area_Chair_qm7S · 2024-12-24

**Metareview:**

(a) This paper proposes  a query-free detector based on transformers to address the limitations of fixed query numbers and detection ambiguity in existing transformer-based detectors. (b) The motivation is clear and the experimental results are convincing. (c) The main weakness lies in the technique contribution and the unclear advantage over fixed query based detector. (d) After carefully reading the paper and reviewers' comments, the AC agrees with the reviewers that the advantage over fixed query based detector is unclear.

**Additional Comments On Reviewer Discussion:**

The authors have provided detailed responses to the reviewers' comments. The responses have addressed reviewer's (rMNn)  concern but failed to address reviewer's (#Pjik) concern about  the advantages of query-free detectors over query-fixed detectors. The other two reviewers do not provide feedback to the authors responses. After reading the paper, I agree with the concern of reviewer (rMNn).

---

### Decision · Program_Chairs · 2025-01-22

Reject